# River suspended-sand flux computation with uncertainty estimation, using water samples and high-resolution ADCP measurements

Jessica Marggraf[1,*], Guillaume Dramais[1,*], Jérôme Le Coz[1], Blaise Calmel[1], Benoît Camenen[1], David J. Topping[2], William Santini[3], Gilles Pierrefeu[4], and François Lauters[5]

[1]RiverLy, INRAE, 5 Rue de la Doua, Villeurbanne, 69100, France
[2]U.S. Geological Survey, Southwest Biological Science Center, Grand Canyon Monitoring and Research Center, 2255 N. Gemini Drive, Flagstaff, AZ 86001, USA
[3]IRD-GET, Institut de Recherche pour le Développement, Laboratoire GET (IRD, CNRS, UPS, CNES), Toulouse, France
[4]CACOH, CNR, 4 Rue de Chalon-sur-Saône, Lyon, 69007, France
[5]Service Etudes Eau Environnement, EDF, 134 Rue de l'étang, Saint Martin Le Vinoux, 38950, France
[*]These authors contributed equally to this work.

**Correspondence:** Dramais Guillaume (guillaume.dramais@inrae.fr), Jessica Marggraf (jessica.laible@web.de)

**Abstract.** Measuring suspended-sand flux in rivers remains a scientific challenge due to their high spatial and temporal variability. To capture the vertical and lateral gradients of concentration in the cross-section, measurements with point samples are performed. However, the uncertainty related to these measurements is rarely evaluated, as few studies of the major sources of error exist. Therefore, the aim of this study is to develop a method determining the cross-sectional sand flux and estimating its
uncertainty. This SDC (for Sand Discharge Computing) method combines suspended-sand concentrations from point samples with ADCP (Acoustic Doppler Current Profiler) high-resolution depth and velocity measurements. The MAP (for Multitransect Averaged Profile) method allows to obtain an average of several ADCP transects on a regular grid, including the unmeasured areas. The suspended-sand concentrations are integrated vertically by fitting a theoretical exponential suspended-sand profile to the data using Bayesian modelling. The lateral integration is based on the water depth as a proxy for the local bed shear stress
to evaluate the bed concentration and sediment diffusion along the river cross-section. The estimation of uncertainty combines ISO standards and semi-empirical methods with a Bayesian approach to estimate the uncertainty due to the vertical integration. The new method is applied to data collected in four rivers under various hydro-sedimentary conditions: the Colorado, Rhône, Isère and Amazon Rivers, with computed flux uncertainties ranging between 18 and 32 %. The relative difference between the suspended-sand flux in 21 cases calculated with the proposed SDC method compared to the International Organization
for Standardization (ISO) 4363 standard method ranges between -40 and +23 %. This method that comes with a flexible, open-source code is the first proposing an applicable uncertainty estimation, that could be adapted to other flux computation methods.

*Copyright statement.* TEXT

## 1 Introduction

The determination of suspended-sediment load is required to estimate sediment dynamics and budgets for river restoration and monitoring, river engineering, and flood risk evaluation (Kondolf et al., 2014). Measuring and monitoring sediment loads and the associated uncertainties within a catchment is a major practical issue for hydrologists and river managers (Hoffmann et al., 2010). Even though suspended-sand transport is a key driver of the river evolution (Kondolf, 1997), it remains difficult to measure its concentration due to its temporal and spatial variability in the cross-section which may account for several orders of magnitude (Armijos et al., 2017). In contrast, fine suspended sediments (<63 $\mu$m) are relatively homogeneous throughout the cross-section, often reaching concentration variations of up to an order of magnitude (Wren et al., 2000).

The total suspended-sediment flux through a cross-section, $\Phi_{\text{total}}$ (kg s$^{-1}$), is defined as the mass of suspended-sediment passing through a river cross-section per unit time:

$$\Phi_{\text{total}} = \int_{y_{\text{lb}}}^{y_{\text{rb}}} q_{\text{ss}}(y)dy = \int_{y_{\text{lb}}}^{y_{\text{rb}}} \int_{z_a}^{h} c(y,z)v(y,z)dzdy \tag{1}$$

where $y$ and $z$ are the lateral and vertical coordinates, $q_{\text{ss}}$ is the suspended-sediment discharge per vertical, $y_{\text{lb}}$ and $y_{\text{rb}}$ are the left and right boundaries of the cross-section, $z_a$ is the reference level for suspension at the top of the bedload layer, generally assumed to be the riverbed elevation, $h$ is the water elevation, and $c(z)$ and $v(z)$ are the time-averaged suspended-sediment concentration and the water velocity perpendicular to the cross-section, respectively.

Suspended-sediment sampling and computing techniques have been developed over decades (Porterfield, 1972; Starosolsky and Rakoczi, 1981; ISO 4363, 2002). Typically, these methods are based on physical water sampling to determine the suspended sediment concentration throughout the cross-section using samples taken at different locations throughout the river cross-section. Samples may be taken following the depth-integrating method, where several nearly complete verticals at different distances from the bank are sampled, or the point-sampling method, where samples are collected at different, discrete water depths and distances from the bank. Different methods were proposed to estimate the suspended-sand flux through the cross-section (Lupker et al., 2011; Shah-Fairbank and Julien, 2015; Santini et al., 2019). The International Organization for Standardization (ISO) standard method (ISO 4363, 2002) consists of computing the velocity-weighted cross-sectional mean concentration by combining physical samples with simultaneous velocity measurements. In this method, the cross-section is divided into $N_{\text{seg}}$ segments and for each increment $l$, the water discharge $Q_l$ and depth-averaged velocity-weighted concentration $C_l$ are evaluated:

$$\Phi_{\text{total}} = \sum_{l=1}^{N_{\text{seg}}} Q_l\, C_l \tag{2}$$

This method is derived directly from the velocity-area method for the measurement of water discharge using current meters (ISO 748, 2009). Even though these classical, discrete methods are widely accepted, they are time- and cost-consuming and sometimes difficult to deploy (Camenen et al., 2023). As they are limited to a few points or depth-averaged samples at a limited number of locations, they are characterized by a low spatial and temporal resolution. Also, the method is directly based

on depth-integrating sampling with no possibility to interpolate and extrapolate results from sampled verticals to the whole cross-section. Some surrogate technologies (e.g. optical and acoustical methods) have been proposed to measure sediment properties and suspended-sand flux with a better spatial and temporal resolution (Wren et al., 2000; Gray and Gartner, 2010).

Acoustic methods using the Acoustic Doppler Current Profilers (ADCP) have become well-established in stream flow monitoring and provide faster, safer and more accurate acquisition of stream velocities, discharges and depths than older current meter methods (Oberg and Mueller, 2007). In a measurement transect, data are acquired on a grid with fixed or variable cell height and many vertical ensembles. For a valid discharge measurement, several cross-sectional transects are typically acquired and processed to obtain information on discharge and velocity. Different post-processing tools have been developed such as the Velocity Mapping Toolbox VMT (Parsons et al., 2013) for the analysis and visualization of cross-sectional velocity data collected along multiple ADCP transects. Other examples are the Discharge Reviews QRev (Mueller, 2016) and QRevInt (Lennermark and Hauet, 2022) developed by an international group, which are applied to ensure discharge measurement reliability and to quantify the uncertainty in the discharge measurement (Despax et al., 2023). While these methods provide temporal averages, that is of each ADCP ensemble, other approaches combine the ADCP measurements spatially close to each other, notably far from the instrument, where the distance between the beams is high (Vermeulen et al., 2014).

In combination with sediment sampling, ADCP measurements of flow velocity and depth can be used to compute the cross-sectional suspended-sand flux (Bouchez et al., 2011; Vauchel et al., 2017). ADCP measurements provide an increased spatial resolution throughout the cross-section compared to point velocity measurements using current meters or rating-curve estimates of the total cross-sectional discharge (Oberg and Mueller, 2007). Moreover, the acoustic backscatter measured by an ADCP, or an Acoustic Backscattering System (ABS) may be used to improve the spatial integration of the concentration in the cross-section. Indeed, the acoustic backscatter can be inverted and used to measure the suspended-sand concentration (e.g. Topping and Wright 2016; Venditti et al. 2016; Szupiany et al. 2019; Vergne et al. 2020). Several software tools have been developed to process ADCP data for estimating suspended-sand flux (Boldt, 2015; Dominguez Ruben et al., 2020) or using backscatter inversions. However, acoustic inversion techniques require many physical samples for calibration, and are affected by acoustic modelling issues (Vergne et al., 2023).

Informed decisions related to sediment monitoring require reliable estimates of the uncertainty of flux measurements. However, the evaluation of the uncertainty is a difficult task because of the complexity of these measurements, due mainly to the temporal and spatial variations of the sediment concentration. Measurement uncertainty is the expression of the statistical dispersion of the values attributed to a measured quantity (JCGM, 2008). Identifying error sources and estimating uncertainty components for suspended-sand measurements have been addressed in many old reports and papers from the FISP (Federal Inter-Agency Sedimentation Project)(FISP, 1941, 1952; Colby, 1964; Guy and Norman, 1970) and more recently by the U.S. Geological Survey (USGS, Topping et al. 2011; Sabol and Topping 2013) and others (Gitto et al., 2017). Moreover, the ISO 4363 (2002) standard proposes a framework to estimate the errors and uncertainty in the mean cross-sectional suspended-sand concentration determined by a point-sampling method. It identifies several sources of error of random and systematic nature. These errors are notably related to the lateral integration of the concentration in between the sampling verticals, thus depending on the number of verticals. Another important error source is the vertical integration of the concentration in between the

sampling points of a vertical, which depends on the number of sampling points along a vertical. Another error is related to the sampling time and consists of the sample's representativity of the natural fluctuations of the concentration due to turbulence. Additional error sources originate from the sampler type and the laboratory analysis. The uncertainty related to each of these error sources is estimated by performing a high number of samples, whereof the average is considered as the approximate true value. This value is taken as reference and the uncertainties originating from the different error sources are estimated based on the deviation to the reference. The respective uncertainty is then determined by the difference between the measured value at a given location and the approximate true value. Even though several sources of error are addressed, the ISO 4363 (2002) method contains several defects in theoretical and practical aspects. First, this standard is not in agreement with the framework proposed in the Guide to the expression of Uncertainty in Measurement (GUM, JCGM (2008)) defining the uncertainty propagation method, notably concerning the notations and the computation of an approximate true value. Second, the large number of additional samples required for the uncertainty analysis and taken under stable hydro-sedimentary conditions is hardly applicable in most environments due to the quick variability of the hydro-sedimentary processes and technical difficulties. For example, to estimate the uncertainty due to the number of verticals (i.e. lateral integration), 15 to 20 verticals with seven point samples each are required for sections less than 100 m wide. The time needed to conduct this kind of survey is practically impossible given the temporal variability of the processes studied. Besides the ISO 4363 (2002) method, no other method proposes a framework addressing all commonly identified sources of error. However, some authors tried to evaluate the main sources of uncertainty. Concerning lateral integration, Colby (1964) noticed that the sand flux varies approximately (within a factor $k_1$) with the third power of the mean velocity $\overline{v}$ for a constant grain size distribution and temperature, and velocities ranging between 0.6 and 1.5 m/s: $\Phi_{\text{total}} = k_1 \overline{v}^3$. Based on these observations and making the required conversions, he stated that the variability of sand concentration at different sampling verticals should be closely related to the variability of $\overline{v}^2/h$, the ratio of the squared mean velocity $\overline{v}$ to the total sampled depth $h$. To ensure comparability among different sampling sections and streams, the $\overline{v}^2/h$ - index, also called $\xi$, may be used :

$$\xi = \frac{\max\left(\overline{v_l}^2/h_l\right)}{\overline{v_z}^2/\overline{h_z}} \tag{3}$$

with $\overline{v_l}$ and $h_l$ the depth-averaged velocity and water-depth for each increment $l$, and $\overline{v_z}$ and $\overline{h_z}$ the depth-averaged velocity and water-depth mean values for the cross-section, respectively. Based on this concept of variability, Guy and Norman (1970) prepared a nomograph that indicates the number of sampling verticals required for a desired maximum acceptable relative standard-uncertainty as a function of the percentage of sand $p_s$ and $\xi$.

The issue of the appropriate sampling time and the associated time-averaging has been been the subject of several studies. Topping et al. (2011) analyzed the temporal variability in sediment concentration among point samples and estimated the associated uncertainties for depth-integrated measurements. In addition, Gitto et al. (2017) concluded that a 9 to 12 minutes sampling time was required to get a representative point sample because of the temporal variability in sediment concentration.

Common methods to estimate the mean cross-sectional suspended-sand concentration and its uncertainty are subject to various limitations such as the interpolation and extrapolation of the suspended-sand concentration towards the river bed and bank, or the impractical feasibility of the ISO 4363 (2002) uncertainty method. Therefore, the first aim of this study is to

introduce a method for computing the total suspended-sand flux with a high spatial resolution. That method combines point samples with ADCP measurements using a physically based understanding of suspended-sediment transport processes. A second aim is to provide a method to estimate the uncertainty related to this suspended-sand flux computation. Therefore, the uncertainties related to several sources of error such as the discharge, the lateral and vertical integration and on the determined point concentrations are estimated and combined following the GUM framework.

First, the proposed Sand Discharge Computing SDC method to estimate the suspended sand concentration and the uncertainty in the sand flux is presented (Sec. 2). Then, this method is applied to four rivers across the world with different flow and sediment characteristics (Sec. 3). Finally, the methodology and presented results are discussed and further developments are suggested in Sec. 4.

We use the following notations:

- $v$ is the water velocity perpendicular to the cross-section.

- $u$ is the absolute standard uncertainty, that is, the standard deviation of the probability distribution of errors,"absolute" meaning expressed in the physical unit of the measurement (e.g., in $m^3 \ s^{-1}$ for discharge);

- $u'$ is the relative standard uncertainty, "relative" meaning expressed in % of the measurement;

- $U = ku$ is the absolute expanded uncertainty, with $k$ a coverage factor taken as $k = 2$, which corresponds to a 95 % probability interval if the distribution of errors is Gaussian;

- $U'$ is the relative expanded uncertainty expressed in % of the measurement result.

## 2 Method

### 2.1 General method

The proposed Sand Discharge Computing SDC method combines ADCP velocity and discharge measurements performed in multiple transects with suspended sand concentrations obtained by point samples distributed throughout the cross-section (Fig. 1). The point sand concentrations are interpolated vertically and laterally in the cross-section using a physically-based method and Bayesian modeling. In the second part of the SDC method, several sources of error are estimated using novel and literature-based approaches and combined to estimate the uncertainty in suspended sand flux measurements.

The SDC method focuses on suspended sand flux measurements in simple river cross-sections without tidal effects or strong secondary currents. In case of strong secondary flow cells causing deviations in the suspended sand concentration from the dominant flow equilibrium, these deviations should be evaluated separately. Choosing an appropriate measurement site is essential to obtain reliable sand flux and uncertainty estimates Edwards and Glysson (1999).

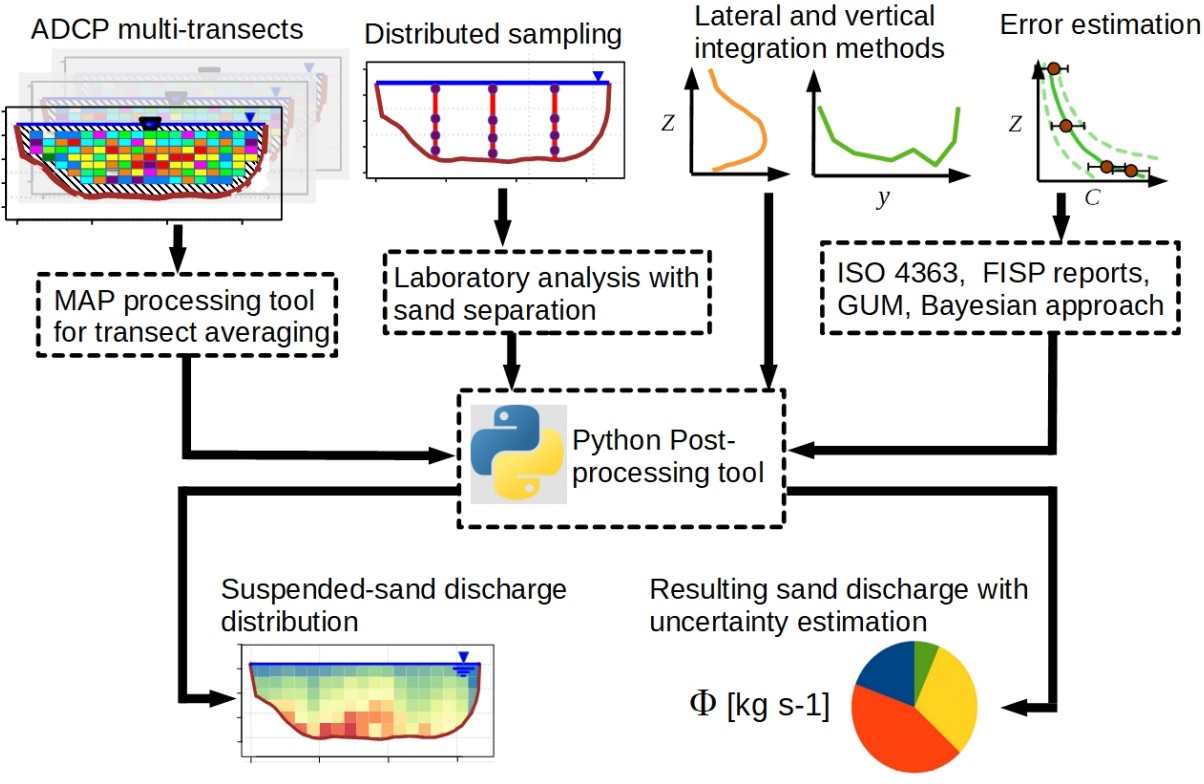

**Figure 1.** Workflow of the proposed Sand Discharge Computing (SDC) method to estimate the suspended-sand flux distribution and the uncertainty in suspended-sand flux through a cross-section.

## 2.2 Physically based method to integrate the concentration in the cross-section

### 2.2.1 Creating an ADCP multitransect averaged profile (MAP)

The Multitransect Averaged Profile (MAP) method was developed to perform the SDC method proposed in thus study (Fig. 1) and implemented in QRevInt (Lennermark and Hauet, 2022) in 2023 (see supplementary material for a detailed description). A typical ADCP discharge measurement consists of the average of individual discharge measurements from successive ADCP transects. The MAP method includes each ADCP transect to generate an averaged transect profile from the bottom to the water surface including the unmeasured areas. This method developed with Python 3 (Van Rossum and Drake, 2009) is based on the QRevInt measurement output. QRevInt provides quality analysis and quality control which allows to have clean input data. For each ADCP discharge measurement, composed of several transects, one averaged MAP profile is computed with a

regular grid for the whole cross-section (including the unmeasured areas). Major differences to similar tools like to the Velocity Mapping Toolbox VMT (Parsons et al., 2013) are that the MAP method can compute an average profile in the absence of a GPS positioning and that it allows the extrapolation of areas unmeasured by the ADCP.

First, the MAP method defines a straight average cross-section on the selected transects of the measurement. Then, these transects are projected using an orthogonal translation on the average cross-section. At this point, each transect is interpolated on the cross-section grid (Fig. 2b). The width and height of cells can be defined by the user. Once each transect is defined on the grid, MAP overlays them to average velocity and depth in each cell of the averaged cross-section (Fig. 2c-e). Finally, velocities are extrapolated to unmeasured areas (Fig. 2f). For edges extrapolation, banks are divided into same shape meshes as the middle streamflow. Mean primary velocity $\overline{v_p}$ (Rozovskii, 1957) on each edge's vertical is computed according power law:

$$\overline{v_p} = \overline{v_{0/1}} \left( \frac{x}{L_{\text{edge}}} \right)^{\frac{1}{m_{\text{edge}}}} \tag{4}$$

where $x$ is the distance of the vertical from the start of the bank, $L_{\text{edge}}$ is the length of the edge, $m_{\text{edge}}$ is an edge-shape exponent (2.41 for a triangular edge, 10 for a rectangular edge, edge extrapolation is not computed otherwise) and $\overline{v_{0/1}}$ is the mean primary velocity from the closest measured vertical. Then, primary velocity on each edges' mesh $v_p$ is computed following a power law with the QRevInt extrapolation exponent $m_{\text{extrap}}$:

$$v_p = \overline{v_p} \frac{m_{\text{extrap}} + 1}{m_{\text{extrap}}} \left( \frac{z_{\text{cell}}}{d} \right)^{\frac{1}{m_{\text{extrap}}}} \tag{5}$$

where $z_{\text{cell}}$ is the depth to centreline of mesh and $d$ the depth on the vertical of the edge. Edges extrapolation of secondary velocity uses a linear law between the closest vertical and the edge. Vertical velocity on the edge follows the distribution of vertical velocities on the closest vertical. MAP generates thus a complete averaged profile with homogeneous cell sizes. Each cell contains information on its distance to the left bank, and its depth and velocity components. Primary and secondary velocities are then transformed into stream-wise and cross-stream velocities in order to compute discharge.

### 2.2.2 Point sampling and laboratory analysis

In addition to the ADCP multi-transects, point samples distributed in the cross-section are performed (Fig. 1). Each suspended-sediment measurement follows the point-sampling method and contains $m$ verticals (typically three to seven) with $N_{\text{sam}}$ samples per vertical (typically four to five). Two types of sampler, a Niskin watertrap-type sampler (instantaneous non-time-averaged sample; Filizola et al. 2009) and isokinetic samplers (US P-06; Spicer 2019) were used in different rivers deployed from boats (Colorado, Rhône and Amazon Rivers) or cable cars (Colorado and Isère Rivers). The target depth is set with a graduated tag line when deploying from the boat or using the depth information from the reel on the cable car. The sampler is equiped with a pressure sensor for post-facto verification (for Rhône, Isère and Amazon Rivers). An electrical valve allows the US P-06 samplers to collect a sample at the desired depth and for the desired sampling duration. For instantaneous samples, taken with the Niskin sampler, a traveler is sent down the rope to close the sampler. For the analysis of the suspended-sediment concentration of each sample, the standard procedure of the American Society for Testing and Materials (ASTM) option C

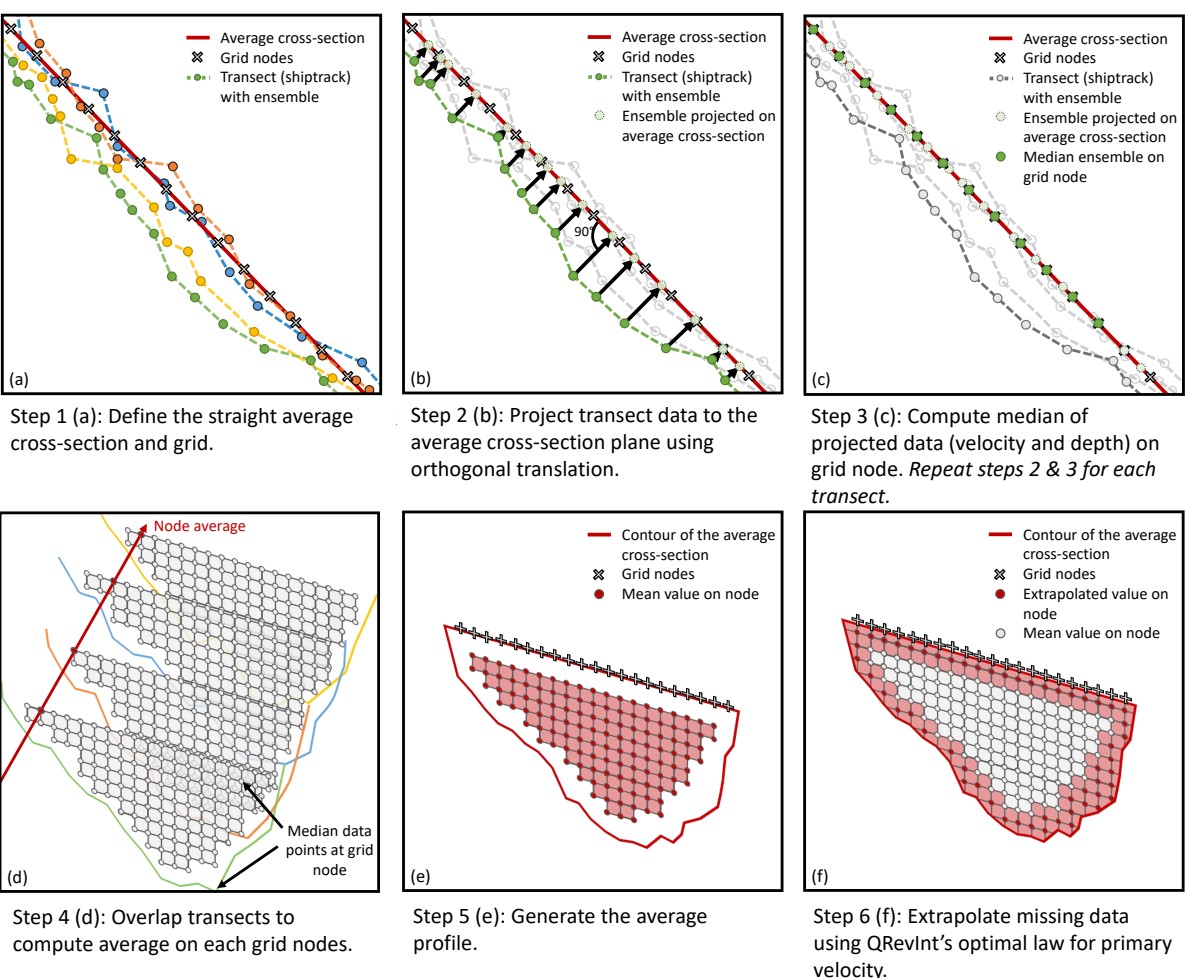

**Figure 2.** Workflow of the transect-averaging procedure deployed by the Multitransect Averaged Profile (MAP) method (based on Parsons et al. (2013)).

(ASTM D3977, 2007) is applied, which consists of separating fine sediments and sand by wet sieving, prior to filtration of the fine sediments.

### 2.2.3 Vertical suspended-sand concentration profiles

A physically based method is applied to assign concentrations to individual cells $(i, j)$ in each sampling vertical $l$ (Fig. 1). It uses a theoretical vertical suspended-sand concentration profile estimated using a Bayesian approach to interpolate and extrapolate the sand concentrations vertically from point samples. The exponential vertical concentration profile proposed by

Camenen and Larson (2008), based on a constant vertical diffusivity $\epsilon_v$ throughout the water column, is defined as:

$$C(z) = C_R \exp(\alpha z) \tag{6}$$

where $C(z)$ (kg m$^{-3}$ or g l$^{-1}$) is the sediment concentration at elevation $z$ above the bed, $\alpha$ is the vertical gradient in a logarithmic scale, and $C_R$ is the bottom reference sediment concentration. Prior to the implementation of the Camenen and Larson (2008) profile, we compared empirically fitted Camenen and Larson (2008) and Rouse (1937) profiles. The differences in sand fluxes are lower than the uncertainty in sand flux of the respective measurement (see supplementary material). Moreover, the profile of Camenen and Larson (2008) has a practical interest that it does not request to arbitrarily define a reference level. Consequently, we used only the Camenen and Larson (2008) profile for the Bayesian modeling in the SDC method. However, other vertical concentration profiles may be used as well and included into the toolbox.

To estimate the concentration profile $C(z)$, the derived depth-averaged concentration and its uncertainty $u'_p$ due to vertical integration (Sec. 2.3.5), the Bayesian Modeling BaM! method is applied (Mansanarez et al., 2019). The BaM! method is based on Bayesian inference, which allows the computation of the posterior probability of a model's parameters from their prior probability and from observations. The model can then be applied to predict the distribution of a new, unobserved data point. The posterior distribution of the parameters is computed using Bayes theorem, and a large number (> 10,000) of realizations are sampled using an adaptive block Metropolis Markov Chain Monte Carlo (MCMC) sampler (Renard et al., 2006) varying the parameters $\alpha$ and $\ln(C(z))$. A linear model is applied using logarithmic concentrations in milligram per liter based on Eq. (6):

$$\ln(C(z)) = \ln(C_R) - |\alpha|z. \tag{7}$$

The first 5,000 realizations are burnt as a warm-up period, then the last 5000 realizations are decimated to decrease the correlation in the results. The MaxPost profile $\ln(C_{n_0}(z))$, the best fitting profile, is computed with the realization of parameters $n_0$ that maximizes the posterior distribution and is used for calculating the sand discharge. From this MaxPost profile, the concentration $C_{i,j}$ in each cell $(i,j)$ along vertical $l$ can be determined. The MaxPost parameters $\ln(C_{R,n_0})$ and $\alpha_{n_0}$ are retained and used for the lateral interpolation (Sec. 2.2.4).

BaM! requires the definition of the prior distribution of the equation parameters, that are here $\ln(C_R)$ and $\alpha$. Both $|\alpha|$ (to assure increasing concentrations with depth) and $C_R$ are strictly positive, therefore they are assumed to follow log-normal distributions with parameters $\mu$ and $\sigma$. Consequently, $\ln(C_R)$ is assumed to follow a Gaussian distribution. The parameters $\mu_\alpha$ and $\sigma_\alpha$ describing the prior distribution of $\alpha$ are the mean and standard deviation of the variable's natural logarithm, respectively. The expected values of $\alpha$ and $C_R$ are evaluated based on local hydro-sedimentary parameters (Camenen and Larson, 2008), which are determined using ADCP depth and velocity, and bedload measurements. The expected value of $C_R$ is calculated using the expression of Camenen and Larson (2008), which is a function of the sedimentological diameter $D_*$, the Shields parameter $\theta$ and the critical bed shear stress $\theta_{cr}$:

$$C_R = 1.5 \, 10^3 \, \theta \, \exp(-0.2 \, D_*) \exp\left(-4.5 \, \frac{\theta_{cr}}{\theta}\right) \tag{8}$$

The sedimentological diameter, or dimensionless grain size, $D_*$ is calculated as:

$$D_* = \overline{D_{50}} \left( \frac{(s-1)\,g}{\nu^2} \right)^{1/3} \tag{9}$$

where $\overline{D_{50}}$ is the median diameter of the sand suspension averaged over the analyzed vertical, $s = 2.65$ is the relative sediment density, $g = 9.81 \text{ m s}^{-2}$ is the acceleration due to gravity and $\nu \approx 10^{-6} \text{ m}^2 \text{ s}^{-1}$ is the kinematic viscosity of water. The expected value of the prior distribution is then converted to $\ln(C_R)$. This reference concentration $C_R$ differs significantly from the reference concentration for a Rouse profile, where the reference concentration is sensitive to the (more or less arbitrary) choice of the reference level adding some complexity to evaluate the priors. Similarly, the expected value of $\alpha$ can be determined as (Camenen and Larson, 2008):

$$\alpha = -\frac{6\,w_s}{\sigma_t\,\kappa\,v_*\,h} \tag{10}$$

where $w_s$ (m s$^{-1}$) is the settling velocity estimated following the formula of Soulsby et al. (1997), $\sigma_t$ is the turbulent Schmidt number, set equal to 1 as a first approximation, $\kappa = 0.41$ is the von Kármán constant, $v_*$ is the total shear velocity and $h$ is the water depth determined using ADCP measurements.

Defining the prior of $\alpha$ as a log-normal ensures that it remains negative under all hydro-sedimentary conditions. This implies that concentration decreases as a function of $z$ away from the bed, thereby corresponding to Rouse mechanics for suspended-sediment computing (Rouse, 1937). The concentration of the finest sizes in suspension may increase away from the bed when the concentration of suspended sediment is relatively high due to 'squeezing' effect or density stratification (Hunt, 1969; McLean, 1992) leading to possible positive $\alpha$-values. We neglect these effects since we focus on sand with relatively low concentrations. Grain size information on vertical $l$ is necessary to determine Eq. (8) and the settling velocity $w_s$, thus to estimate both $\ln(C_R)$ and $\alpha$. In case they are not available, no prior parameters are defined and the model is fitted by BaM! on the observations only.

The second parameter $\sigma_\alpha$ of the log-normal distribution of $\alpha$ can be estimated by uncertainty propagation equations established following the Guide to the expression of Uncertainty in Measurement (GUM, JCGM (2008)). It is estimated based on the relative uncertainty $u'_\alpha$ of $\alpha$, supposing $\sigma_\alpha = u'_\alpha$. This approximation works well for small values ($< 0.5$) of $\sigma$ of the respective log-normal distribution. The parameter $\sigma_\alpha = u'_\alpha$ is estimated by propagation from Eq. (10):

$$u'_\alpha = \sqrt{u'^2_{ws} + u'^2_{\sigma t} + u'^2_\kappa + u'^2_{v*} + u'^2_h}, \tag{11}$$

Only a few studies evaluated these uncertainties. Since additional experimental measurements are beyond the scope of this article, we define, based on literature, the uncertainty in the settling velocity $u'_{ws} = 5\%$ (Camenen, 2007), the uncertainty in the turbulent Schmidt number $u'_{\sigma t} = 20\%$ (Gualtieri et al., 2017), $u'_\kappa = 0$ (theoretical value with negligible variations Smart (2022)), the uncertainty in the shear velocity $u'_{v*} = 5\%$ (Perret et al., 2023) and the uncertainty in the elevation of the sampled point within the water column $u'_h = 5\%$ (Dramais, 2020). With these values, we obtain $u'_\alpha = \sqrt{0.0475} \approx 21.8\%$.

The second parameter $\sigma_{\ln(C_R)}$, the standard deviation of $\ln(C_R)$, could also be determined by an uncertainty propagation derived from the Data Reduction Equation of $C_R$ (Eq. (8)). However, it has been shown that the highest uncertainty is related

to the structural uncertainty of the formula of $C_R$ itself, not to its parametric uncertainty (Camenen et al., 2014). Indeed, the dataset used to establish the semi-empirical formula of Camenen and Larson (2008) is characterized by a large scatter, with differences of about 50 % between the measured and predicted concentration (other formulas also come with large structural uncertainty). Consequently, it is assumed that $\sigma_{\ln(C_R)} = u'_{C_R} = 50\%$.

### 2.2.4 Lateral interpolation

The lateral interpolation of the suspended-sand concentration to calculate $C_{i,j}$ in every cell of the MAP-grid is based on a physical approach using the water depth as an index (Fig. 1). Following Camenen and Larson (2008), $C_R$ is set proportional to the local bed shear stress, which can be assumed to be proportional to the water depth $h$, if the friction slope is constant throughout the river cross-section (Khodashenas and Paquier, 1999; Camenen et al., 2011). Thus, the ratio $C_{R,j}/h_j$ of the reference concentration $C_R$ and the water depth $h$ for each column $j$ in the MAP-grid is estimated through linear interpolation along the cross-section. As a first approximation, $\alpha_j$ is assumed independent of the local bed shear stress, since it is mostly influenced by large scale turbulence structures (Van Rijn, 1984). $\alpha_j$ varies linearly with horizontal distance between two adjacent sampling verticals (where $\alpha$ was estimated from the concentration profiles fitted to the samples), and remains constant between the first/last sampling vertical and the edge of the cross-section.

### 2.2.5 Determination of concentration $C_{i,j}$ in each MAP-cell $(i,j)$

The proposed SDC method is based on the discretization of the river cross-section by a regular grid fitted on the ADCP data (MAP-grid) composed of $N_j$ columns and $N_i$ depth cells (Fig. 1). The general idea is to assign a concentration and discharge to each cell, so that a flux per cell can be obtained after multiplication. The total cross-sectional sand flux $\Phi_{\text{total}}$ is calculated by summing up the suspended sand fluxes per cell:

$$\Phi_{\text{total}} = \sum_{i=1}^{N_i} \sum_{j=1}^{N_j} \Phi_{i,j} \tag{12}$$

where $\Phi_{i,j}$ (kg s$^{-1}$) is the suspended-sand flux through one MAP cell $i,j$. The suspended-sand flux $\Phi_{i,j}$ can be calculated as:

$$\Phi_{i,j} = C_{i,j}Q_{i,j} = C_{i,j}u_{i,j}w_j h_{\text{cell,i,j}} \tag{13}$$

where $C_{i,j}$ (kg m$^{-3}$ or g l$^{-1}$) and $Q_{i,j}$ (m$^3$ s$^{-1}$) are the suspended-sand concentration and liquid discharge through each cell $(i,j)$, respectively, $u_{i,j}$ (m s$^{-1}$) is the normal velocity component, $w_j$ (m) is the width, $h_{\text{cell,i,j}}$ (m) is the height of the $i^{th}$ vertical cell in the $j^{th}$ column in the MAP-grid. The discharge $Q_{i,j}$ through each cell is determined using the novel MAP-method based on QRevInt (Lennermark and Hauet, 2022) and the suspended-sand concentration $C_{i,j}$ is determined following the novel, physically based SDC method.

The parameters $\alpha$ and $C_R$ are evaluated for each MAP-cell $(i,j)$ applying the presented vertical and lateral integration. The suspended-sand concentration in each cell in the MAP grid is thus evaluated as:

$$C_{i,j} = \frac{1}{h} \int\limits_{z_{i,j}-h_{\text{cell,i,j}}/2}^{z_{i,j}+h_{\text{cell,i,j}}/2} C_{R,j} \exp(\alpha_j\, z) dz \qquad (14)$$

## 2.3 Estimation of the uncertainty in measurements of suspended-sand flux through a cross-section

### 290 2.3.1 General method

The uncertainty $U'_\Phi$ in measurements of the suspended-sand flux through a cross-section is based on the calculation of suspended-sand flux (i.e. Eq. (2), Fig. 1). Therefore, the flux $\Phi$ is the product of discharge $Q$ and mean cross-sectional concentration $\overline{C}$: $\Phi = Q \times \overline{C}$. Thus, $U'_\Phi$ can be separated into a factor related to discharge $U'_Q$ and one related to the concentration $U'_C$:

$$295 \quad U'_\Phi = \sqrt{U'_Q{}^2 + U'_C{}^2} \qquad (15)$$

Equation (15) is based on the hypothesis that the errors in discharge and concentration are independent, otherwise, the term has to include the associated covariances. Such assumption that the errors are independent appears however reasonable. First, because discharge and concentration are measured independently, the discharge is measured using ADCPs and the concentration is determined using sampling and laboratory analyses. Second, the error sources in discharge and concentration are significantly different. Velocity lateral interpolation errors are negligible due to the high spatial resolution of ADCP measurements whereas concentration lateral interpolation errors are large. We agree that these two error components are physically correlated, but this is not a problem as the first one is negligible.

To approximate the uncertainty $U'_\Phi$ in the suspended-sand flux through a cross-section, both uncertainties $U'_Q$ and $U'_C$ on the discharge and concentration have to be determined (Fig. 3).

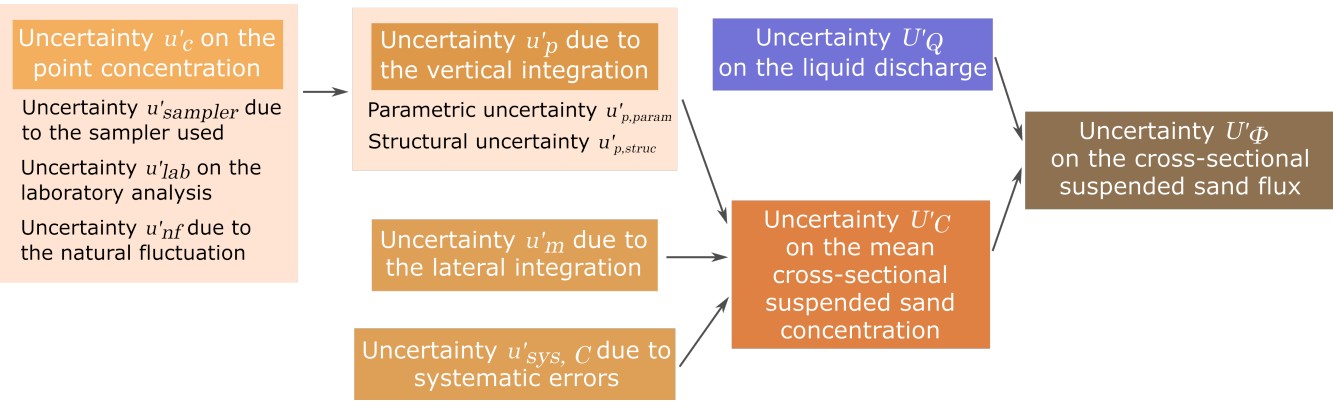

**Figure 3.** Flow diagram of the developed approach to estimate the uncertainty $U'_\Phi$ in the cross-sectional suspended-sand flux.

The uncertainty $U'_Q$ in multiple-transect ADCP discharge measurements is calculated following the OURSIN-method (Despax et al., 2023) as implemented in the open-source software QRevInt (Lennermark and Hauet, 2022). $U'_C$ is the combination of several uncertainty components (cf. Fig. 3) listed in Table 1 and detailed afterwards.

### 2.3.2 Uncertainty $U'_C$ in the mean cross-sectional suspended-sand concentration

The uncertainty $u'^2_C$ in the mean cross-sectional suspended-sand concentration is calculated as:

$$u'^2_C = u'^2_{\text{sys,C}} + u'^2_m + \sum_{l=1}^{m} \frac{\Phi_l^2}{\Phi^2} u'^2_{p,l}, \tag{16}$$

where $u'_{\text{sys,C}}$ is the uncertainty due to systematic errors in the concentration, $u'_m$ is the uncertainty due to the lateral integration based on the number $m$ of verticals and $u'_{p,l}$ is the total uncertainty due to the vertical integration estimated for each vertical $l$ (cf. Fig. 3) and $\Phi_l$ is the suspended sediment flux through vertical $l$.

### 2.3.3 Uncertainty due to systematic sources of error $u'_{\text{sys,C}}$

Following the ISO 4363 (2002) method, the uncertainty due to systematic sources of error $u_{\text{sys,C}}$ is expressed as:

$$u'^2_{\text{sys,C}} = u'^2_{\text{sys,m}} + u'^2_{\text{sys,p}} + u'^2_{\text{sys,lab}} + u'^2_{\text{sys,sampler}}, \tag{17}$$

where $u'_{\text{sys,m}}$ is the uncertainty due to the systematic error of the flux computation scheme, $u'_{\text{sys,p}}$ is the uncertainty due to the systematic error of the vertical integration, $u'_{\text{sys,lab}}$ is the uncertainty due to the systematic error of the laboratory analysis and $u'_{\text{sys,sampler}}$ is the uncertainty due to the systematic error of the sampler type, since the underlying errors are assumed systematic. These terms, detailed in ISO 4363 (2002), remain constant, independently of the increasing number of sampling points or verticals (cf. Table 1).

### 2.3.4 Uncertainty $u'_m$ due to lateral integration

To facilitate the application compared to the standardized approach (ISO 4363, 2002), the uncertainty $u'_m$ due to lateral integration, is estimated based on Eq. (3) and the nomograph published by Guy and Norman (1970):

$$u'_m = 0.4\, p_s\, (1.43\, \xi - 1.37)\, m^{-0.7}, \tag{18}$$

with $p_s$ the percentage of sand in the suspension, $\xi$ (cf. Eq. (3)) the $\overline{v}^2/h$-index (Colby, 1964) and $m$ the number of verticals. If only the suspended-sand flux through a cross-section is measured or is of principal interest, as in our study, the percentage of sand in the equation should be supposed to be 100 %, neglecting the influence of the fine-sediment flux. This does not signify that no suspended fine sediments are present, but allows to take account of the considerable lateral gradients of the sand suspension. It also avoids underestimating the uncertainty due to the lateral integration, because the uncertainty $u'_m$ for the same sediment discharge measurement (same $m$ and $\xi$) is higher, when assuming $p_s = 1$ than including fine sediments. This approach is applied in our study, although the lateral interpolation applied differs slightly, as it is based on the water depth $h$

**Table 1.** Uncertainty components used to compute the uncertainty $U'_C$ on the mean cross-sectional suspended sand concentration.

| Error sources | Notation (standard uncertainty) | Nature | Estimation method |
|---|---|---|---|
| Lateral interpolation | $u_{\mathrm{m}}$ | Systematic | Nomograph from Guy and Norman (1970) |
| Vertical interpolation | $u_{\mathrm{p}}$ | Random | Bayesian approach (Sec. 2.3.5): total uncertainty combining the next two components |
| Vertical interpolation | $u_{\mathrm{p,param}}$ | Random | Bayesian approach (Sec. 2.3.5): parametric uncertainty |
| Vertical interpolation | $u_{\mathrm{p,struc}}$ | Random | Bayesian approach (Sec. 2.3.5): structural uncertainty |
| Concentration measurements | $u_{\mathrm{meas}}$ | Random | Formula (Eq. (19)) combining the following three components |
| Sampler type | $u_{\mathrm{sampler}}$ | Random | Fixed values ($8\%^{1}$ and $16\%$) |
| Laboratory analysis | $u_{\mathrm{lab}}$ | Random | Formula (Eq. (20), Gordon (2000)) |
| Natural fluctuations of concentration | $u_{\mathrm{nf}}$ | Random | Repeated measures experiments (Sec. 2.3.9) |
| Systematic concentration errors | $u_{\mathrm{sys,C}}$ | Systematic | Formula (Eq. (17)$^{1}$) combining the four next components ($3.54\%$) |
| Systematic lateral integration errors | $u_{\mathrm{sys,m}}$ | Systematic | Fixed value ($1.5\%^{1}$) |
| Systematic vertical integration errors | $u_{\mathrm{sys,p}}$ | Systematic | Fixed value ($2\%^{1}$) |
| Systematic laboratory analysis errors | $u_{\mathrm{sys,lab}}$ | Systematic | Fixed value ($2\%^{1}$) |
| Systematic sampler type errors | $u_{\mathrm{sys,sampler}}$ | Systematic | Fixed value ($1.5\%^{1}$) |

[1] Value originating from ISO 4363 (2002)

and the parameters $C_R$ and $\alpha$ of the vertical profiles. However, this approach is assumed to be consistent with our modified lateral interpolation.

### 2.3.5 Uncertainty $u'_p$ due to vertical integration

The uncertainty $u'_p$ is determined for each vertical $l$ from the distribution of vertically integrated concentrations computed from the profiles estimated by the Bayesian approach described in Sec. 2.2.3. This uncertainty accounts for the uncertainty $u'_{\text{meas}}$ (estimated in Sec. 2.3.6) in point concentrations taken as observational data in the Bayesian inference. The integration of the previously obtained vertical concentration profiles $\ln(C_n(z))$ (Sec. 2.2.3) allows the determination of the parametric uncertainty $u'_{p,\text{param}}$. However, computing the total uncertainty $u'_p$ due to vertical integration requires the inclusion of structural errors at the elevation of the sampling points prior to the vertical integration. These structural errors are representative of the residuals between the point measurements and the exponential profiles. The structural uncertainty can be estimated from the total uncertainty $u'_p$ and the parametric uncertainty $u'_{p,\text{param}}$: $u'_{p,\text{struc}}$ as $u'^2_{p,\text{struc}} = u'^2_p - u'^2_{p,\text{param}}$.

The parametric uncertainty $u'_{p,\text{param}}$ can be determined from the distribution of concentration profiles $\ln(C_n(z))$ computed in Sec. 2.2.3 (Fig. 4a). Each of these $n$ profiles is converted to $C_n(z)$ (Fig. 4b) and linearly interpolated applying a trapezoidal integration to determine its depth-averaged concentration $\overline{C_n}$, which is converted to $\ln(\overline{C_n})$. Application of the entire procedure for all simulations $n$, then yields a distribution of depth-averaged concentrations $\ln(\overline{C_n})$. The mean value of this distribution is $\overline{\ln(\overline{C})}$ and the standard deviation is the uncertainty $u'_{p,\text{param}}$ based on the assumption $\sigma = u'_{p,\text{param}}$ (Fig. 4c).

To determine the structural error, the prior distribution of its standard deviation is defined as log-normally distributed with $\mu = 0$ and $\sigma = 1$ in the BaM! method. For every sampling point at the elevation $z$, a normally distributed error with mean zero and standard deviation $u'_{\text{meas}}$ is defined. An error is then drawn from this distribution and added to the estimated concentration $\ln(C_n(z))$ for every simulation $n$ to obtain a modified vertical profile $\ln(C_{\text{mod},n}(z))$ (Fig. 4d). In the next step, the same procedure as for the estimation of $u'_{p,\text{param}}$ is applied: conversion of $\ln(C_{\text{mod},n}(z))$ to $C_{\text{mod},n}(z)$, vertical averaging to obtain $\overline{C_{\text{mod},n}}$ and conversion to $\ln(\overline{C_{\text{mod},n}})$. The mean value of the resulting distribution is the mean depth-averaged concentration $\overline{\ln(\overline{C_{\text{mod}}})}$ and its standard deviation is the total uncertainty $u'_p$ due to vertical integration, based on the assumption $\sigma = u'_p$ (Fig. 4f).

### 2.3.6 Uncertainty $u'_{\text{meas}}$ in point concentrations

As point concentration errors are accounted for in the Bayesian analysis of vertical concentration profiles, the uncertainty $u'_{\text{meas}}$ in point concentrations is already included in the uncertainty $u'_p$ due to vertical integration. Therefore, in contrast to the ISO 4363 (2002) method, u'meas does not explicitly appears in Eq. (16). The uncertainty $u'_{\text{meas}}$ is calculated as:

$$u'_{\text{meas}} = \sqrt{u'^2_{\text{sampler}} + u'^2_{\text{lab}} + u'^2_{\text{nf}}}, \tag{19}$$

where $u'_{\text{sampler}}$ is the uncertainty due to the sampler type, $u'_{\text{lab}}$ is the uncertainty due to the laboratory analysis and $u'_{\text{nf}}$ is the uncertainty due to natural fluctuations in sediment concentration arising from turbulence (Fig. 3).

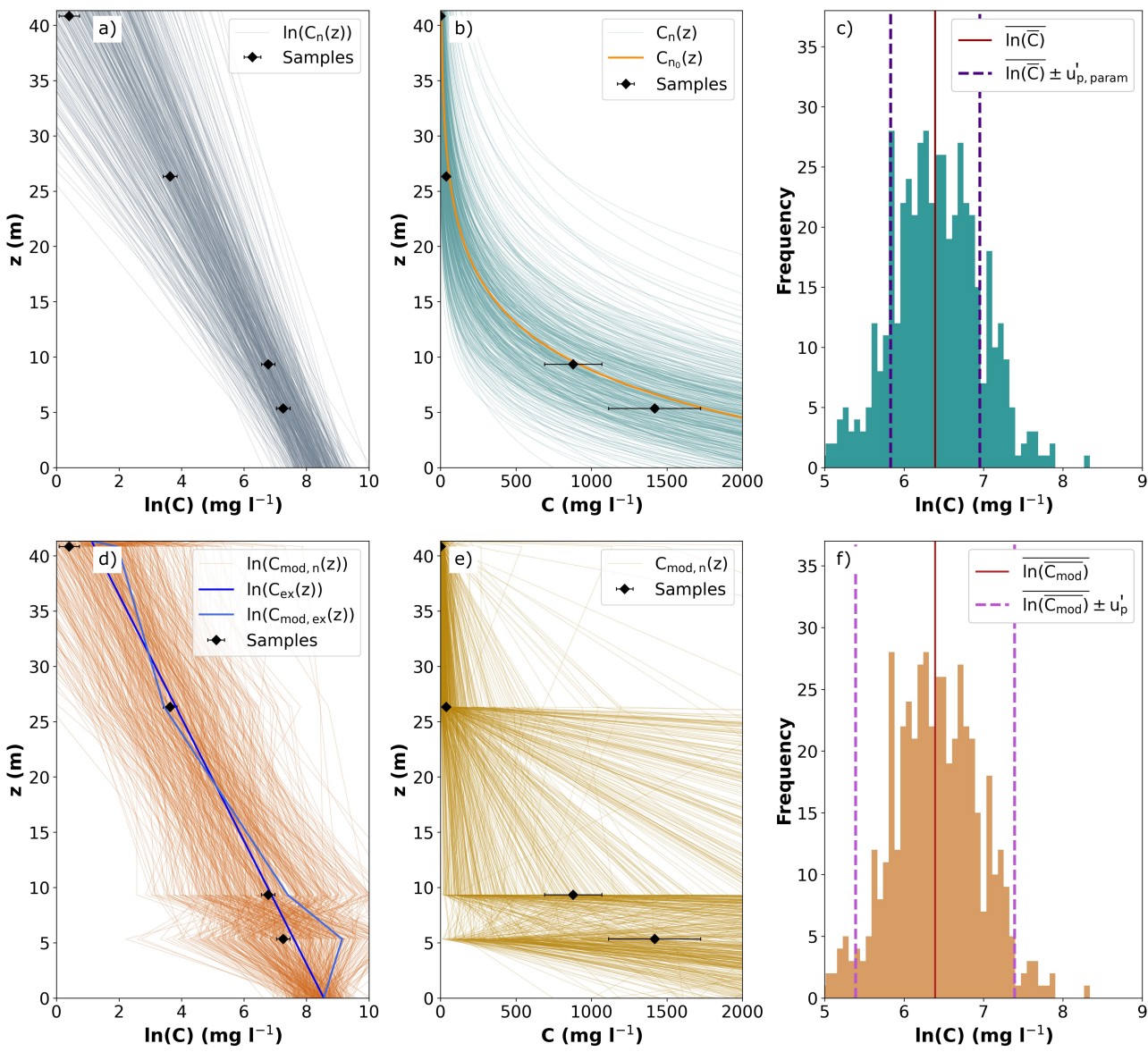

**Figure 4.** Workflow for the estimation of the uncertainty $u'_p$ due to vertical integration, including the estimation of the parametric uncertainty $u'_{p,\mathrm{param}}$ in **(a,b,c)** and of the total uncertainty $u'_p$ in **(d,e,f)**. **(a,d)** Vertical concentration profiles $\ln(C_n(z))$ and $\ln(C_{\mathrm{mod,n}}(z))$, respectively, sampled through Bayesian interference and including the structural error and two exemplary profiles in **(d)**. **(b,e)** Vertical concentration profiles $C_n(z) = \exp(\ln(C_n(z)))$ and $C_{\mathrm{mod,n}}(z) = \exp(\ln(C_{\mathrm{mod,n}}(z)))$, respectively, with the MaxPost profile $C_{n_0}(z)$ in **(b)**. **(c,f)** Histograms of depth-averaged concentrations $\ln(\overline{C_n})$ and $\ln(\overline{C_{\mathrm{mod,n}}})$ with the mean depth-averaged concentrations $\overline{\ln(\overline{C})}$ and $\overline{\ln(\overline{C_{\mathrm{mod}}})}$ as well as the standard deviations $u'_{p,\mathrm{param}}$ and $u'_p$, respectively.

### 2.3.7 Uncertainty $u'_{\text{sampler}}$ due to the sampler type

Even though several comparisons have been conducted, the distribution of random errors related to a specific sampler type is difficult to assess. For example, a review of the values of $u'_{\text{sampler}}$ used in different studies is provided by Dramais (2020). In this study, the value suggested in the ISO 4363 (2002) standard is used for isokinetic samplers such as the US P6: $u'_{\text{sampler}} = 8\ \%$. To account for the greater uncertainty arising from non-isokinetic sampling, this uncertainty is arbitrarily doubled for non-isokinetic samplers: $u'_{\text{sampler}} = 16\ \%$.

### 370 2.3.8 Uncertainty $u'_{\text{lab}}$ due to laboratory analysis

Many studies have estimated the random uncertainty related to the measurement of (fine) sediment concentration in the laboratory (e.g. by filtration). The ISO method estimates an uncertainty of 1.5 % due to the random error and an uncertainty of 2 % due to the systematic error (ISO 4363, 2002). Based on an intercomparison study of different laboratories, Gordon et al. (2000) determined a standard uncertainty for the fine and sand fractions separately. We use the approach of Gordon et al. (2000) at the

375 68 % confidence level and a given sand concentration $C$ (g l$^{-1}$) in the analysed sample:

$$u'_{\text{lab}} = 1.091\, C^{-0.5}. \tag{20}$$

### 2.3.9 Uncertainty $u'_{\text{nf}}$ due to natural fluctuations

To approximate the uncertainty $u'_{\text{nf}}$ due to the natural fluctuations in concentration and grain size in the point samples arising from turbulence, a simplified method, similar to the ISO method or the "At-a-Point-Error" (APE) Topping et al. (2011) is

380 applied. To this end, several points are repeated at different hydro-sedimentary conditions with a time difference of less than 1 hour between the first and last sample and the suspended-sand concentration $C_i$ is calculated for each sample. One sampling point is repeated three to nine times and the mean sediment concentration $\overline{C_{\text{rep}}}$ of the respective set of measurements is determined. Based on the nomenclature of the ISO 4363 (2002), this mean concentration $\overline{C_{\text{rep}}}$ per set can be understood as the "approximate true value". The relative standard deviation $u'_{\text{rep}}$ for each set of $N_{\text{rep}}$ repetitions is then calculated following ISO

4363 (2002):

$$u'_{\text{rep}} = \sqrt{\frac{\sum_{i=1}^{N_{\text{rep}}} \left(\frac{C_i}{\overline{C_{\text{rep}}}} - 1\right)^2}{N_{\text{rep}} - 1}} \tag{21}$$

Performing this calculation for all repetitions, the relative uncertainty for each set of repetitions $u'_{\text{rep}}$ can be plotted versus the mean concentration $\overline{C_{\text{rep}}}$ per set (Fig. 5).

The number of sets of repetitions and tested hydro-sedimentary conditions within this study is limited compared to the variety

of sampling conditions. In the best case, these measurements should be conducted on every sampling campaign, however, in reality, hardly possible using the presented measurement techniques. Using the same sampling protocol, the sampling campaign

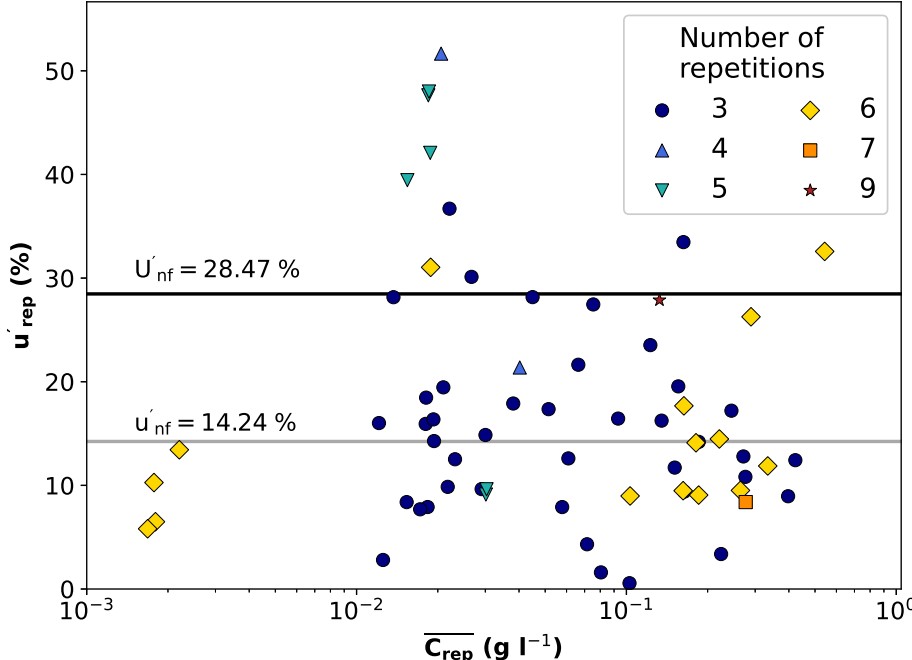

**Figure 5.** The relative uncertainty $u'_{\text{rep}}$ and mean concentration $\overline{C_{\text{rep}}}$ for each set of repetitions (these repetitions include data from Isère, Colorado, Toutle, and Cowlitz Rivers (Spicer, 2019)). The relative uncertainty $u'_{\text{nf}}$ and the relative expanded uncertainty $U'_{\text{nf}}$ due to natural fluctuations correspond to the average and the average multiplied with a coverage factor of $k = 2$ of all tested relative uncertainties $u'_{\text{rep}}$.

with many additional samples for the uncertainty estimation would take very long, so that the variation in river discharge would become too great. Therefore, a constant uncertainty $u'_{\text{nf}} = 14.24\,\%$ is determined based on these results and applied to all point measurements, which corresponds to the median of all tested relative uncertainty $u'_{\text{rep}}$. The enlarged uncertainty of

395 $U'_{\text{nf}} = 28.47\,\%$ at a 95 % confidence interval corresponds roughly to the estimations made by Gitto et al. (2017) in the Fraser River (they found 3 to 33 % of uncertainty range for individual 30 s samples). Furthermore, it should be noted that only a small range of hydro-sedimentary conditions at a given sampling location is sampled by this empirical approach. The uncertainties are probably higher than estimated here and may as well be grain-size dependent (Topping et al., 2011).

## 3 Application

### 3.1 Survey sites

The proposed SDC method was applied to four datasets from different rivers around the world. Each dataset includes suspended-sediment measurements following the above presented protocol and ADCP data.

### 3.1.1 Rhône River

The Rhône River is one of the major rivers of Europe, heading from the Rhône Glacier in the Alps, and running through western Switzerland and south-eastern France. Mostly a gravel-bed river, it is the largest silt and clay contributor to the Mediterranean sea (Delile et al., 2020). The presented measurements were conducted near the gaging station (V3000015) at Lyon Perrache (WGS84 coordinates: 45.742344, 4.826738), France, where the Rhône River drains a catchment of about 20 300 km$^2$ with a mean annual discharge of about 600 m$^3$ s$^{-1}$ (Dramais, 2020).

### 3.1.2 Isère River

The Isère River is an Alpine river and the largest tributary of the Rhône River by suspended sediment flux (Poulier et al., 2019). At Grenoble, France (WGS84 coordinates: 45.197747, 5.768566) , where the measurements were conducted (gaging station W1410010) (Némery et al., 2013), the mean annual discharge is about 180 m$^3$ s$^{-1}$ with a catchment area of 5 700 km$^2$.

### 3.1.3 Colorado River

The Colorado River is one of the most iconic rivers in the western United States. The measurements took place at the U.S. Geological Survey (USGS) Colorado River above Little Colorado River near Desert View (WGS84 coordinates: 36.203484, -111.800917), Arizona gaging station at the River Mile 61. This station (number 09383100, U.S. Geological Survey 2023) has a mean annual discharge of 306 m$^3$ s$^{-1}$ and a catchment area of 296,000 km$^2$. Suspended-sediments are monitored since a long time in this area (Topping et al., 2021).

### 3.1.4 Amazon River

The Amazon River basin exceeds 6 000 000 km$^2$ in area. The Amazon River is the largest river in the world by discharge. The Manacapuru gauging station (14100000) is part of the Critical Zone Observatory HyBAm (Hydrology of the Amazon Basin) and operated by the French National Research Institute for Sustainable Development (IRD), the Brazilian National Agency (ANA), and the Brazilian Geological Service (CPRM). This station has been used for more than 40 years by the Brazilian national hydrometric network to provide data on the Amazon (Solimões) River (WGS84 coordinates: -3.324377, -60.561183). At this station, the Amazon River watershed is approximately $2\times10^6$ km$^2$ and average water discharge is about 103 000 m$^3$ s$^{-1}$ (Filizola et al., 2009).

Those four survey sites, with various geomorphological conditions (see Table 2), were sampled according to the above-described ADCP-measurement and point-sampling procedures. The Isère River was sampled with an isokinetic US P-06 sampler and the other rivers with watertrap-type sampler (Niskin).

**Table 2.** Hydraulic and morphological characteristics of the survey sites. $Q_w$ is the total ADCP water discharge including extrapolations areas, $Q_{Meas.}$ is the ADCP measured water discharge (excluding extrapolations areas), $\bar{v}$ the mean cross-sectional velocity, the aspect ratio is the river width divided by mean depth, $D_{50}$ is the median diameter of suspended-sand, $p_s$ the percentage of sand in the suspension, $\xi$ is an index relating stream velocity and depth, and $U'_\Phi$ is the expanded uncertainty in measurements of suspended sand flux at 95% confidence level.

| River and location | Survey date | | $Q_w$ (m³ s⁻¹) | $Q_{Meas.}/Q_w$ (%) | $\bar{v}$ (m s⁻¹) | $W$ (m) | $H$ (m) | Aspect ratio (-) | Sand $D_{50}$ (μm) | $p_s$ (%) | $\xi$ (-) | $U'_\Phi$ (%) |
|---|---|---|---|---|---|---|---|---|---|---|---|---|
| Rhône at Lyon Perrache (France) | Jan. | 22th 2018 | 2000 | 81 | 1.2 | 170 | 12 | 14 | 100-300 | 26.9 | 1.07 | 18.6 |
| Isère at Grenoble (France) | Apr. | 6th 2022 | 120 | 52 | 1.1 | 70 | 3 | 23 | 90-290 | 46.6 | 1.23 | 19.1 |
| Colorado River Mile 61 (USA) | Feb. | 19th 2019 | 370 | 65 | 1.1 | 100 | 5 | 20 | 100-130 | 68.2 | 1.6 | 31.1 |
| Amazon at Manacapuru (Brazil) | Apr. | 19th 2012 | 144000 | 83 | 1.6 | 3400 | 43 | 79 | 180 | 76.6 | 1.33 | 26.6 |

## 3.2 Vertical suspended-sand concentration and flux profiles

Measured suspended-sand point concentrations are fitted with an exponential profile to extrapolate the concentrations to the unmeasured parts of the water column and also interpolate between points. There is substantial vertical and lateral variability in suspended-sand concentration at all study sites (Fig. 6). Indeed, different vertical gradients $\alpha$ and/or reference concentrations $C_R$ are observed among the measured verticals. The highest sand concentrations and largest gradients, with a difference of up to three orders of magnitude between the bottom and surface concentrations, are observed in the Amazon River (cf. Fig. 6d). This indicates a possible effect of the water depth on the concentration gradient through the vertical diffusion coefficient (inversely proportional to the water depth, Eq. (10)) and/or due to dunes (whose height is typically proportional to the water depth, and so can enhance the shear velocity). In contrast, the concentrations at the other sites range between 0.01 and 0.5 g $l^{-1}$. In the various surveys, sand-concentration gradients are associated with particle-size gradients, with coarser particles closer to the riverbed. The measured concentrations vary strongly at some verticals, so that they do often not correspond to the fitted vertical concentration profiles, not even when taking the uncertainty $U'_{\mathrm{meas}}$ on the point concentrations into account. This uncertainty usually varies for the presented samples between 20 and 25 % at a 95 % confidence interval.

Vertical profiles of suspended-sand flux (Fig. 7) are determined by multiplying the suspended-sand concentration in each cell in the MAP-grid with the discharge in the same cell. Similarly, the point suspended-sand fluxes are the product of the point concentration and the discharge of the surrounding cell in the MAP-grid. Consequently, decreasing fluxes close to the bed, as expected based on theory, are hardly or not at all visible for most verticals sampled in the Isère (Fig. 7b) or the Colorado (Fig. 7c). Large differences between the point fluxes and the profiles result notably from poorly fitted vertical concentration profiles, e.g. vertical 34 in the Isère River or vertical 1650 in the Amazon River. In other words, when the point concentrations do not follow an exponential profile, there are large differences between point fluxes and profiles.

## 3.3 Suspended-sand flux through a cross-section

The suspended-sand concentration in each MAP cell $(i, j)$ is calculated by applying the lateral interpolation and extrapolation of the profile coefficients $C_R$ and $\alpha$ (Sec. 2.2.3). The spatial view of the cross-sections highlights the distribution of the suspended-sand concentration (Fig. 8). Different layers in some measurements appear due to the vertical and horizontal resolution of the ADCP data, i.e. the size of the MAP cells. As the vertical integration is based on the water depth, the lateral interpolation of the profile coefficients produces high concentrations near the bed, especially when there are large water depth variations and when vertical measurements are made on the deepest parts. This is clearly observed Fig. 8d in between the central and right sampling vertical on the Amazon.

The mean cross-sectional suspended-sand concentrations $\overline{C_{\mathrm{SDC}}}$ and fluxes $\Phi_{\mathrm{SDC}}$ computed with the SDC method, are compared to the ISO method (ISO 4363, 2002) using the relative differences $\epsilon_C = (C_{\mathrm{SDC}} - C_{\mathrm{ISO}})/C_{\mathrm{ISO}}$ and $\epsilon_\Phi$, respectively. The results for the suspended sand concentration are in close agreement between the two methods, with $\epsilon_C$ ranging between -2 and 3.5 % for three examples (Table 3), whereas a significant concentration difference is observed between both methods for the Colorado computations (-15.8 %). The most likely hypothesis to explain this difference is that the surface sample at the

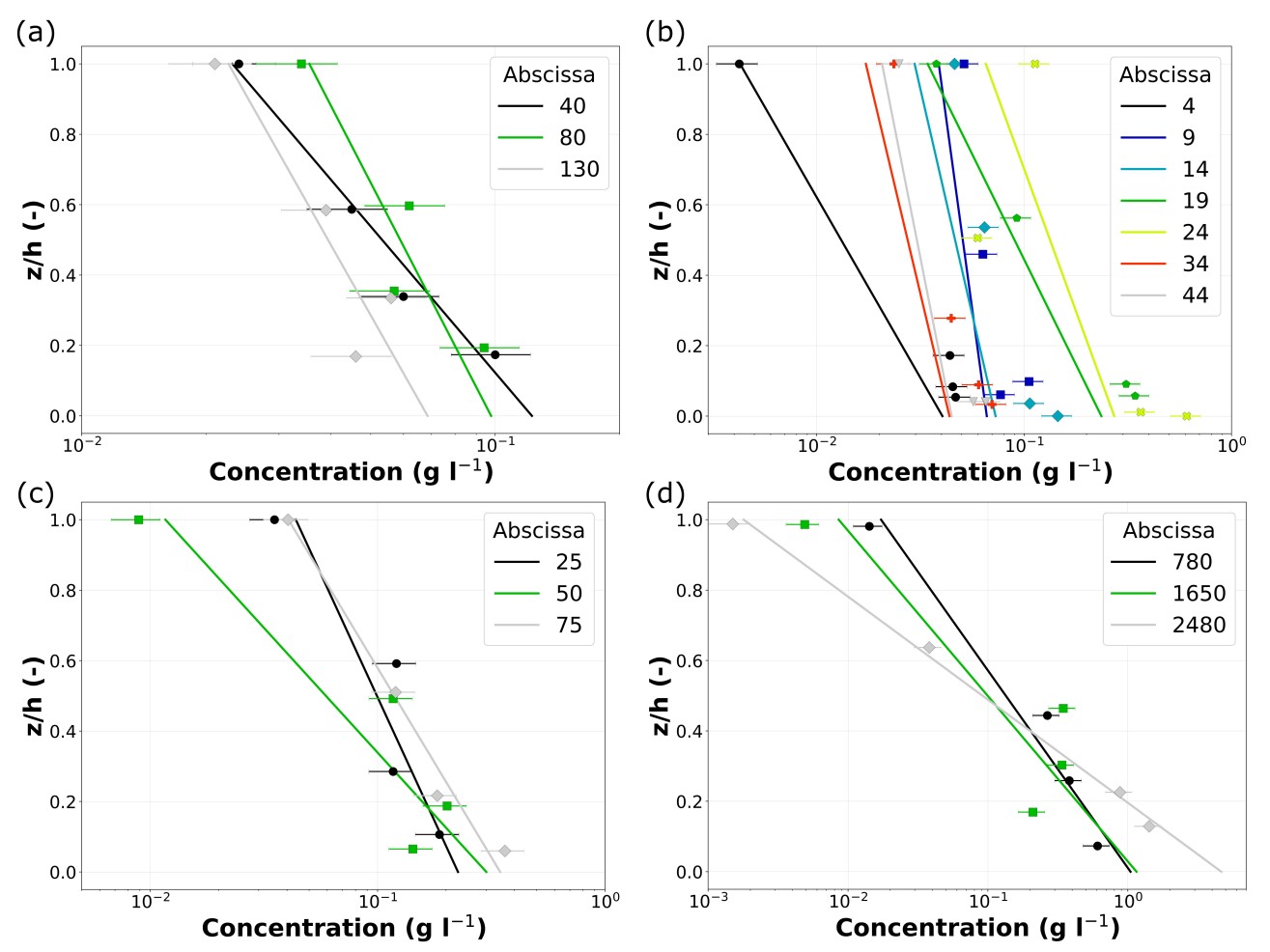

**Figure 6.** Measured sand concentrations with uncertainty $U'_{\text{meas}}$ and exponential fits (represented as colored lines) for each sampling vertical using Bayesian modelling for the **(a)** Rhône River at Lyon Perrache, **(b)** Isère River at Grenoble Campus, **(c)** Colorado River at River Mile 61, and **(d)** Amazon River at Manacapuru. The colors correspond to the sampling verticals, which are indicated by their distance in meters from the left bank, called abscissa.

middle of the transect has a relatively low sand concentration (abscissa 50, Fig. 6c). This low concentration heavily influenced the fit of the vertical profile and reduced the flux in this part of the cross-section, which is the place of the most intense flow. This highlights one of the limitations of the method when only a few points are used for suspended-sand flux computation in the cross-section.

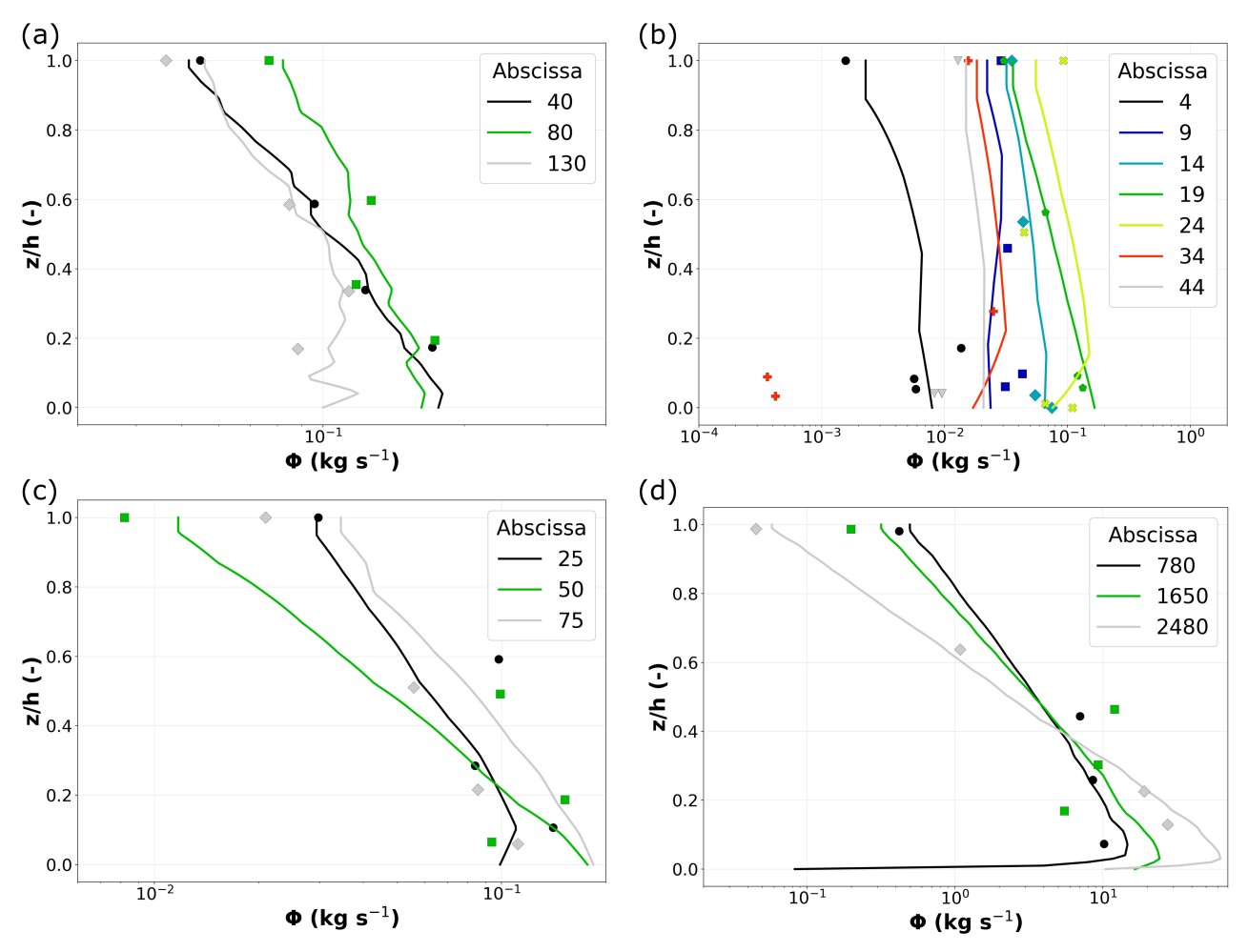

**Figure 7.** Suspended-sand fluxes interpolated in each MAP cell along the sampling verticals (lines) and measured at the sampling points for the **(a)** Rhône River at Lyon Perrache, **(b)** Isère at Grenoble, **(c)** Colorado River at River Mile 61, and **(d)** Amazon River at Manacapuru. The colors correspond to the sampling verticals, which are indicated by their distance in meters from the left bank, called abscissa.

**Table 3.** Mean cross-sectional suspended-sand concentrations and total fluxes for the four presented measurements using the ISO method and SDC method.

| Study site | $\overline{C_{\text{ISO}}}$ (g l$^{-1}$) | $\overline{C_{\text{SDC}}}$ (g l$^{-1}$) | $\epsilon_C$ (%) | $\Phi_{\text{ISO}}$ (kg s$^{-1}$) | $\Phi_{\text{SDC}}$ (kg s$^{-1}$) | $\epsilon_\Phi$ (%) |
|---|---|---|---|---|---|---|
| Rhône River | 0.051 | 0.050 | -2.0 | 102 | 99.3 | -2.8 |
| Isère River | 0.085 | 0.088 | 3.5 | 9.2 | 9.5 | 2.9 |
| Colorado River | 0.120 | 0.101 | -15.8 | 48.1 | 40.5 | -16 |
| Amazon River | 0.304 | 0.298 | -2.0 | 40284 | 39438 | -2.1 |

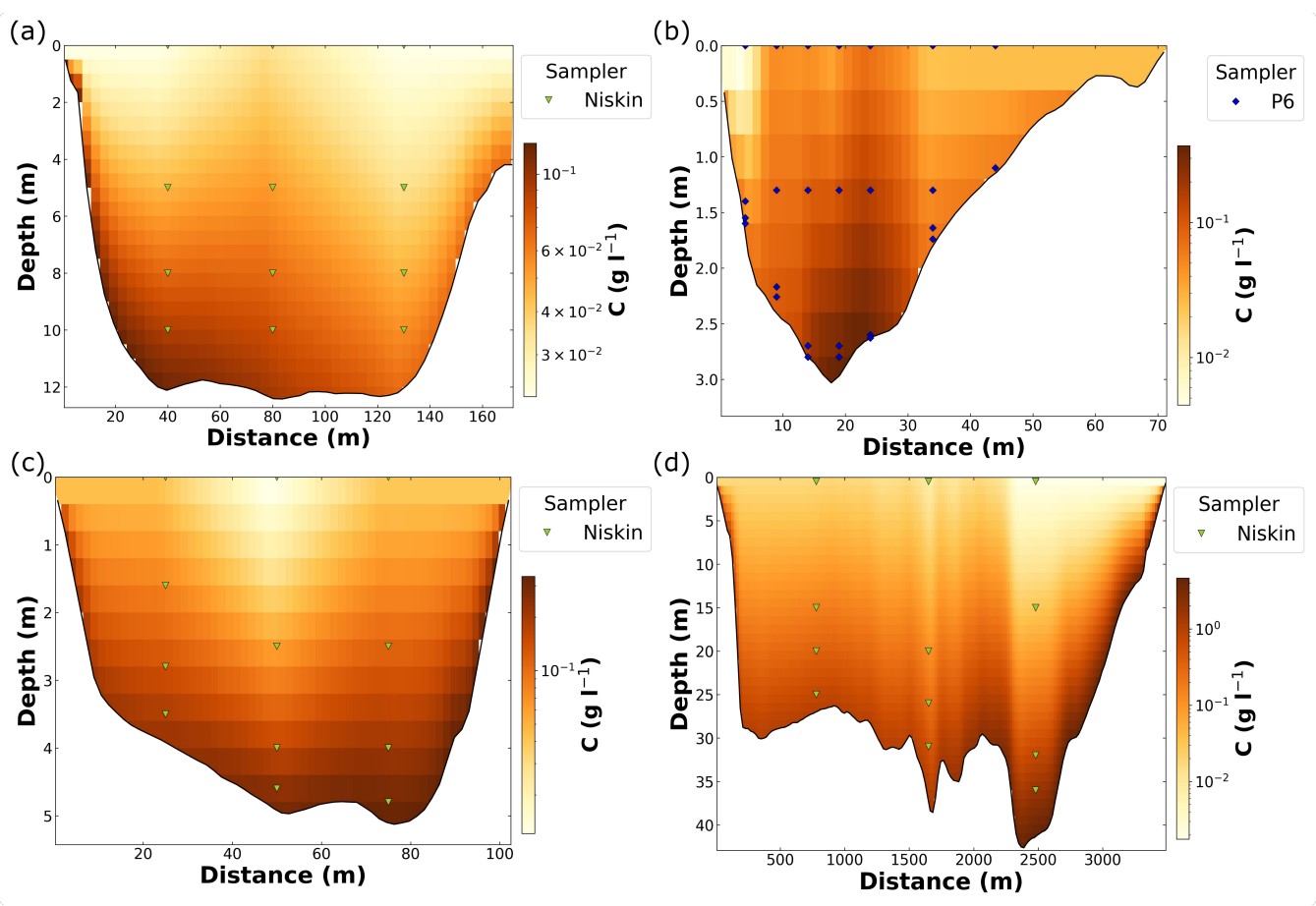

**Figure 8.** Suspended-sand concentrations calculated with SDC method in each cell of the MAP grid throughout the cross-section and location of the sampling points of the **(a)** Rhône River at Lyon Perrache, **(b)** the Isère River at Grenoble Campus, **(c)** the Colorado at River Mile 61, and **(d)** the Amazon River at Manacapuru (d).

### 3.4 Suspended-sand uncertainty evaluation

The total uncertainty and the contribution of each error source to total variance are evaluated for the four measurements (Fig. 9). The absolute uncertainty at a 95 % confidence interval with a coverage factor of $k = 2$ ranges between 19 % and 31 %. The main uncertainty components are the uncertainty $U_p'$ due to vertical integration and the uncertainty $U_m'$ due to lateral integration. The uncertainty $U_p'$ is displayed as its two components, the parametric uncertainty $u_{p,\mathrm{param}}'$ and the structural uncertainty $u_{p,\mathrm{struc}}'$. The parametric uncertainty $u_{p,\mathrm{param}}'$ is determined by the information from the priors and from the sampling points used to calibrate the model, particularly their number, distribution along the vertical, and uncertainty $u_{\mathrm{meas}}'$. Increasing the number of samples and decreasing the uncertainty $u_{\mathrm{meas}}'$ would decrease this uncertainty $u_{p,\mathrm{param}}'$. In contrast, the structural uncertainty

$u'_{p,\text{struc}}$ is estimated from the residuals of the fit of the model to the calibration points. The farther they are from the fitted vertical concentration profile and the lower their uncertainty, the greater is the uncertainty $u'_{p,\text{struc}}$.

The uncertainty $U'_m$ is estimated from index $\xi$, representing the lateral homogeneity of the cross-section in terms of depth and discharge with the percentage of sand in the suspension and the number of sampling verticals. However, as only sand concentrations are considered here, the percentage of sand was set equal to 100 %. At relatively high $\xi$-values, such as on the Colorado River, a larger number of verticals would have been required to decrease the uncertainty $U'_m$, whereas this uncertainty is relatively low in a more uniform river like the Rhône River with a low $\xi$-value. In that latter case, three verticals are sufficient to describe the lateral distribution of the concentration in the cross-section. The measurements on the Isère and Amazon Rivers are both characterized by similar $\xi$. The large number of sampling verticals in the Isère River (seven) leads to a low uncertainty $U'_m$, whereas the uncertainty and small number of sampling verticals in the Amazon River (three) leads to high uncertainty $U'_m$. Nevertheless, it has to be taken into account that the lateral integration is based on the water depth, whereas the calculation of the uncertainty $U'_m$ is based on $\xi$. These two different ways to conduct the lateral integration of the suspended-sand concentration and the uncertainty estimation may affect the results. The contributions of both the uncertainties $U'_Q$ on the liquid discharge and the systematic uncertainties $u'_{\text{sys}}$ to the uncertainty $U'_\Phi$ on the suspended-sand flux are typically low.

## 4 Discussion

### 4.1 A physically based method

The novel SDC method offers a number of advances for cross-sectional sediment flux measurement, especially the physically based integration of concentration.

For the vertical integration of concentration, the first step of the method is based on the fit of an exponential profile. The uncertainty related to the fit of this profile is estimated. As discussed in Camenen and Larson (2007), very similar results would have been obtained by fitting a Rouse profile (see Fig. 1 and 2 in the supplementary material). However, another more detailed approach (Hunt, 1969; McLean, 1992) may provide a better fit and thus modify the results. Consequently, various theoretical approaches including the effects of suspended-sediment stratification or due to size distribution of sediment may be integrated into the toolbox, allowing to choose the best fitting semi-empirical model to the sampling conditions. Currently, only the exponential profile and the Rouse profile (not presented here, but included in the code) are available in the toolbox.

The goodness of the fit, thus the structural uncertainty, depends not only on the chosen profile, but also on the number and position of the sampling points. Dramais (2020) applied the same Bayesian approach and showed that the points measured close to the bed have a great influence on the reference concentration $C_R$ value and on the slope $\alpha$ of the fitted exponential profile. In contrast, subsurface samples may in some cases bias the fit of the exponential profile. Interestingly, the best compromise was observed when sampling points are positioned close to the bed and for a number of five samples per profile.

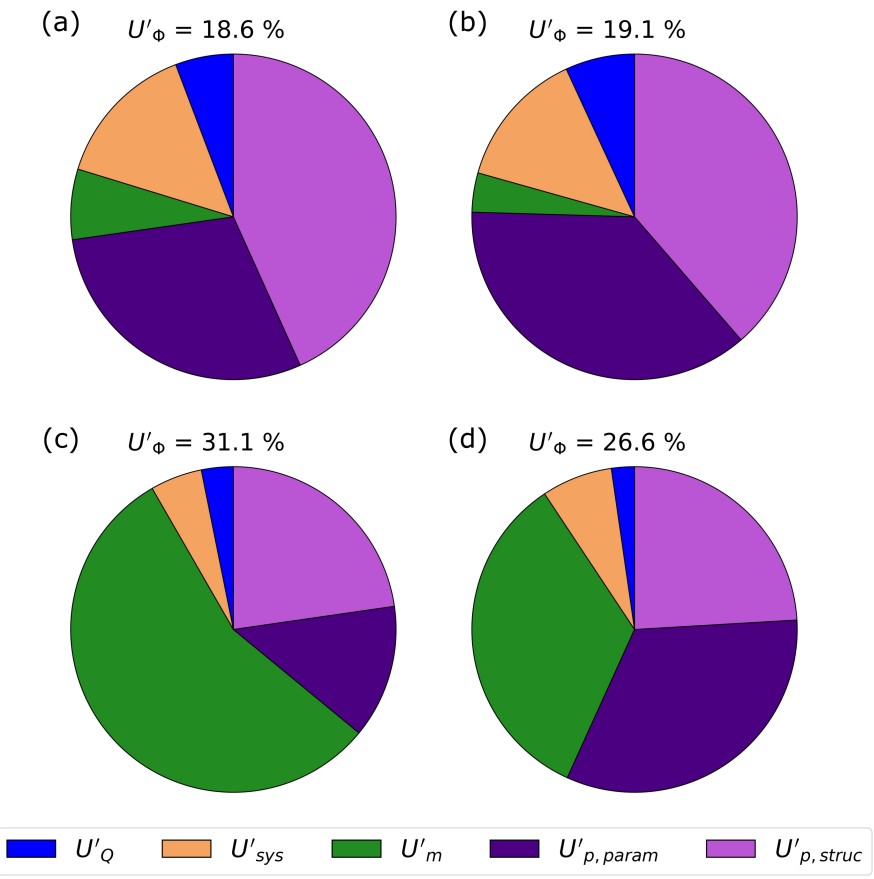

**Figure 9.** Total expanded uncertainty $U'_\Phi$ in suspended sand flux and the relative contributions of uncertainty components to the total variance on the **(a)** Rhône River at Lyon Perrache, **(b)** on the Isère River at Grenoble, **(c)** on the Colorado at River Mile 61, and **(d)** on the Amazon River at Manacapuru. The five components are the expanded uncertainty $U'_Q$ in the liquid discharge, the expanded uncertainty $U'_{\mathrm{sys}}$ due to systematic sources of error, the expanded uncertainty $U'_m$ due to the lateral integration of the concentration, the expanded parametric uncertainty $U'_{\mathrm{param}}$ due to the vertical integration and the expanded structural uncertainty $U'_{\mathrm{struc}}$ due to the vertical integration.

Depth-integrating measurements could be a solution to avoid errors due to the vertical interpolation which occur with fitting point samples. However with this protocol, lateral extrapolations and extrapolations in the unmeasured parts of the cross-section are not possible with a physical base. Depth-integrating samples could then be associated with a larger uncertainty.

The lateral integration of the concentration may be improved as well, even though the SDC method presents an advance by using the water depth as a proxy for the shear stress. The current method leads to artefacts of relatively high concentrations close to the river bed within extrapolated areas (Fig. 8d). Moreover, the use of the parameters of the vertical concentration profile for the lateral integration increases the importance of a well-fitted vertical profile, otherwise, large uncertainties may result.

A major advance in both the vertical, but particularly the lateral integration, may be made using the acoustic backscatter measured by the ADCP. One option is the use of the backscattered signal intensity as a proxy for suspended-sediment concentration facilitating the inter- and extrapolation in between the sampling points. Moreover, several studies (Bouchez et al., 2011; Venditti et al., 2016; Szupiany et al., 2019) and commercial softwares (e.g. ASET (Dominguez Ruben et al., 2020)) focus on the vertical moving-boat backscatter inversion to gain information on suspended-sediment. For the inversion of single-frequency applications as the ADCP, strong assumptions or calibrations are necessary to estimate correctly the concentration and grain size of silt-clay and sand-sized sediment (Vergne et al., 2023). Additionally, the issue of unmeasured areas close to the river bed, surface and banks persists and requires the extrapolation of the estimated concentrations, e.g. by applying theoretical suspended sand transport formulas (Dominguez Ruben et al., 2020). However, when using the ADCP backscatter for the calculation of the concentrations, the errors in discharge and concentration would probably not be independent anymore, because the same measurement method is used and the uncertainty due to the spatial integration of the concentration, the major error source of the uncertainty in concentration, will decrease. Consequently, the formulation of the uncertainty estimation should be adapted, e.g. by including the covariances.

## 4.2 A high-resolution ADCP data based method

Our study proposes a general method which uses high-resolution ADCP data from successive transects. Compared to existing multi-transect averaging tools, the newly developed ADCP multi-transect averaged profile (MAP) provides an average dataset including the unmeasured areas. MAP uses RDI or SonTek raw binary files, and reduces the pre-processing error, as it uses data quality filters from QRevint. The method may use either the bottom track or the GPS as reference and the user can customize vertical and lateral dimensions of the resulting grid cells. The obtained regular grid then facilitates the further analysis steps.

One limitation of the SDC method is due to the vertical and lateral integration of concentrations which can be limited by the ADCP data resolution. The vertical and lateral integration of concentration are evaluated on each MAP grid cell. Although this approach allows the estimation of concentrations close to the bed and banks, the size of the grid cells is limited by the size of the ADCP cells. Consequently, if the ADCP spatial resolution is low, the resulting mean concentration may be affected. An example of this problem is provided by the data from the Colorado River (Fig. 10), where cells were set with a height of approximately 0.4 m (Fig. 10a) and 0.8 m (Fig. 10b). In some cases, it could be meaningful to adapt the ADCP cells size to increase the resolution of the measurement and consequently increase the resolution of the resulting cross-sectional estimation of the distribution of the sediment flux (e.g. Vermeulen et al. (2014)).

## 4.3 A method open to various sampling protocols

Another advantage of the SDC method is its suitability to different point sampling protocols, with various numbers and locations of sampling points along the verticals and varying numbers of verticals. This flexibility is particularly useful, when specific areas or depths are of major interest requiring more detailed sampling or if the sampling points are not distributed in the cross-section following the ISO protocols. Also, it provides an estimation of the suspended-sand concentration close to the bed or banks, in areas excluded by many methods such as the ISO method.

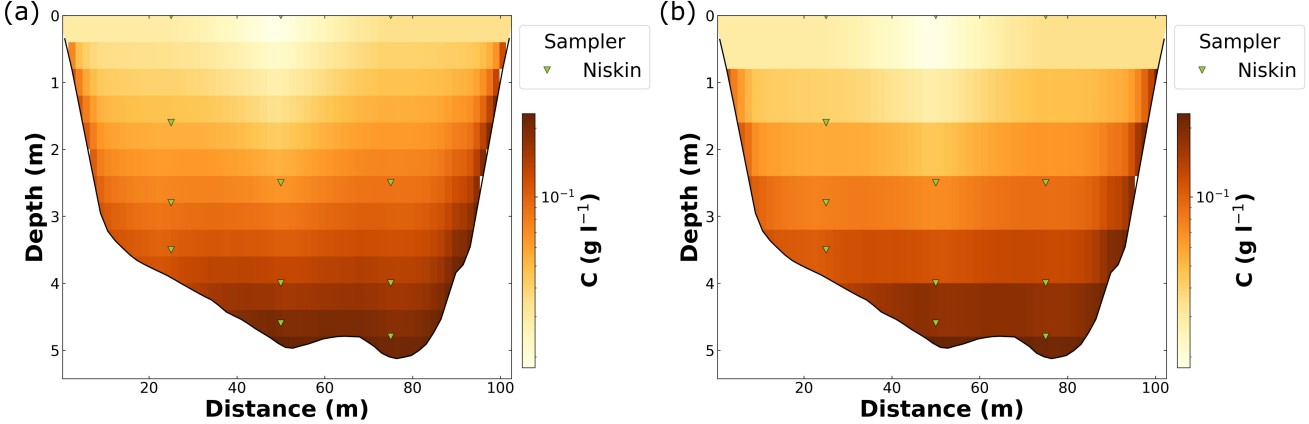

**Figure 10.** Influence of the vertical cell height on the cross-sectional suspended-sediment measurement in the Colorado River at River Mile 61, **(a)** with a cell height of 0.4 m and **(b)** with a cell height of 0.8 m.

We evaluate the differences of the suspended-sand flux calculated using the proposed SDC method relative to the flux calculated using the ISO method using a larger dataset for Rhône River and Isère River that encompasses the detailed examples shown above. For this supplementary data, conditions were different but they were at the same locations. The relative difference $\epsilon_\Phi$ ranges for all four studied rivers between -40 % and +23 % with no clear relationship with the total flux (Fig. 11). These values are lower than the estimated uncertainties, underlining the importance of the errors associated to the sand flux measurement. Similar observations can be made when using an empirically fitted Rouse profile instead of the exponential profile fitted with Bayesian modeling (see supplementary material). In comparison, typical ADCP water discharge measurements are characterized by an uncertainty of 5 - 12 % (results from several repeated measures experiments performed in France (Despax et al., 2019)).

The relative differences between the sand fluxes calculated following the SDC method compared to the ISO method may reach up to 40 %, but no bias is visible. Extensive field and laboratory measurements under stable and known conditions are required to evaluate the performance of the two methods and determine the best among these two and other existing methods. Divergences may originate from the difference in the sampling protocols. The ISO method is based on a sampling protocol with seven points per vertical, where the samples are taken at precise relative depths, while four samples are typically taken per vertical and at varying relative depths in the present measurements. In addition, the SDC method may be more accurate than the ISO method at high discharges or deep water depths, when it is difficult to lower the sampler close to the river bed. However, accurate flux references are lacking in rivers, so that this assumption cannot be verified experimentally.

## 4.4 A first estimation of the suspended-sand flux uncertainty

This method combines existing and novel approaches to estimate the uncertainty in the suspended-sand flux taking all error sources into account and their contributions in the uncertainty budget, which represents a major advance over the existing ISO

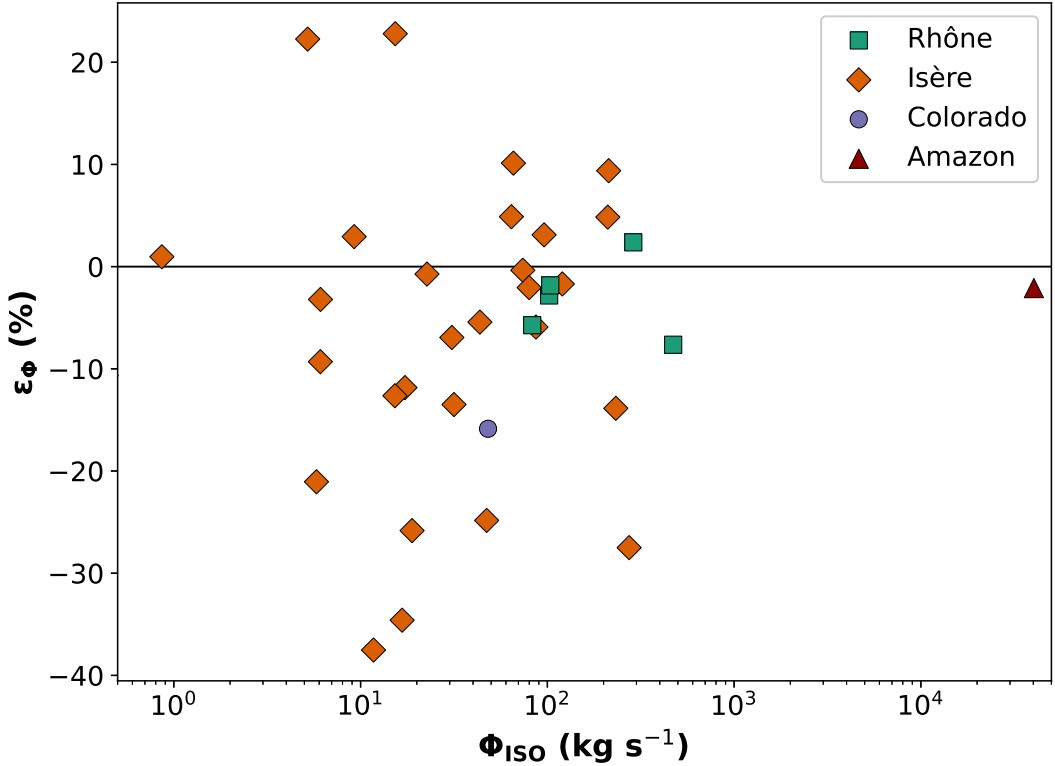

**Figure 11.** Relative difference $\epsilon_\Phi = (\Phi_{\mathrm{SDC}} - \Phi_{\mathrm{ISO}})/\Phi_{\mathrm{ISO}}$ as a function of the suspended-sand flux $\Phi_{\mathrm{ISO}}$ determined using the ISO method for all four studied rivers, $\Phi_{\mathrm{SDC}}$ is determined using the SDC method for all four studied rivers.

method. The method is easy and fast to apply and contains no empirical calculation, except for the uncertainty $u'_{\mathrm{nf}}$ on natural concentration fluctuations. Compared to the ISO method, this reduces considerably the required sampling time and effort,

facilitating its application even for monitoring purposes. The price for this is additional assumptions on the spatial distribution of concentrations, based on simple physical basis valid for gauging cross-sections with a simple flow. Our method allows to interpolate and extrapolate concentrations based on a physically based approach. Furthermore, the introduced Bayesian approach we introduced to compute $u'_{\mathrm{p}}$ appears to be a promising way to analyze vertical concentration profiles and the related uncertainties.

This new approach does not have only advantages, and certain limitations have already been identified. Here are some suggestions for correcting them or improving the code. Concerning the uncertainty estimation, major advances could be made by developing a more robust method for the estimation of the uncertainty $u'_m$ due to the lateral integration, the uncertainty $u'_{\mathrm{sampler}}$ due to the sampler and the uncertainty $u'_{\mathrm{nf}}$ due to the natural fluctuations. The major issue about the estimation of the uncertainty $u'_m$ is its difference to the physical approach followed for the lateral integration of the concentration. The

determination of the $\overline{v}^2/h$-index, $\xi$, accounts for the water depth and the stream velocity, whereas only the water depth is

taken into account for the lateral interpolation of the concentration. Developing a consistent method for both uses and adapted to various channel geometries and lateral concentration gradients is needed, e.g. following approaches developed for index-velocity relations (e.g. Kästner et al. (2018)). Similarly, robust methods to estimate the uncertainty $u'_{\text{sampler}}$ due to the sampler and the uncertainty $u'_{\text{nf}}$ due to natural fluctuations for different settings should be developed. Finally, potentially uncovered error sources such as the uncertainty in the vertical position and the uncertainty related to the total sampling duration (Topping et al., 2011; Gitto et al., 2017) should be estimated and integrated as well. Moreover, the estimation of the prior distributions of $\alpha$ and $\ln(C_R)$ to estimate the uncertainty $u'_p$ due to the vertical integration could be estimated as well. Other vertical concentration profiles may be included, and the best fitting formula may be chosen limiting thereby the uncertainty in the vertical integration.

## 4.5   An open source method

A fully operational and open-source toolbox is available. This toolbox includes several options not presented in detail in this article, like the use of an empirically fitted Rouse profile. The code is relatively flexible, suitable for various conditions and protocols commonly applied in point sampling protocols. It allows the computation of suspended-sand, but also silt-clay concentration if available for various sampler types and deployment conditions. If silt-clay concentrations are available, the ratio of silt-clay concentration to sand concentration can be compared at each sampling point and also over the entire cross-section. The vertical position of the sampler may be determined as well as the transit time between the water surface and the final sampling depth if pressure-sensor measurements are available. If available, the results of the flux measurements may be related to data from adjacent hydro-sedimentary gauging stations. If grain-size data are available for several samples, they are visualized and a mean cross-sectional grain size distribution following ISO 4363 (2002) is calculated.

## 5   Conclusion

The new SDC method presented in this study allows meaningful determination of the suspended-sand flux through a river cross-section with uncertainty. Therefore, this method merges data from ADCP discharge measurements and point suspended-sediment samples. The SDC method includes a method for averaging several ADCP transects with discharge and velocity measurements on a regular grid in the entire cross-section is developed. Suspended-sand concentrations obtained by point sampling are then vertically interpolated by fitting a physical-based exponential concentration profile and choosing the best fit using a Bayesian framework (BaM!). The lateral interpolation between the point samples and extrapolation in the unmeasured zones are performed on a physical basis. Both the vertical and lateral integrations allow the computation of the suspended-sand concentration for each ADCP grid cell and consequently the suspended-sand flux.

The toolbox presented in this article proposes a major advance in the estimation of the uncertainty in point suspended-sand sampling. It addresses several sources of error and integrates existing methods with novel approaches to propose an applicable framework. The main error sources are identified as $u_m$ due to lateral integration and $u_p$ due to vertical integration, thereby justifying the SDC method, which seeks to improve spatial integration in the whole cross-section.

The application of the methodology on several cross-sectional suspended-sand measurements conducted following different sampling protocols on four global rivers yields results that slightly differ from the ISO method (-15.9 % to +2.9 % suspended-sand flux difference). This approach can be easily used and is adaptable to different sampling cases; the only requirement is an ADCP discharge measurement including several transects and a point samples dataset. The data processing, analysis and visualization toolbox is open-access and available online.

Future development may benefit by focus on the incorporation of the acoustic backscatter measured by the ADCP to guide the vertical and lateral integration, and on the development of more robust methods of estimating the uncertainties due to lateral integration, the sampler performance and the natural fluctuations in concentration arising from turbulence.

## Author contribution

GD, JL, JLC, BC conceptualized this work. The methodology was developed by GD, JL, JLC and BC. The code and uncertainty part was wrote by JL and BCal. GD and JL wrote the original draft of this paper, which was edited by DT, JLC, BCal and BC. JLC and BC supervised the work. All authors brought ideas into the work, participated on the field measurements and reviewed the final draft.

## Data availability

Data are available on: https://doi.org/10.57745/NLFT7Q and https://doi.org/10.57745/YTCYSX.

## Code availability

The open-source toolbox is available at: https://gitlab.irstea.fr/jessica.laible/analysis-solid-gauging.

## Acknowledgements

The authors thank Thibault Vassor and Alexis Pavaux for their help in this project and Janna Stepanian (INRAE) for her help in the sample analysis. Thanks to K. Spicer (USGS) for sharing data. Thanks to Dr. Benjamin Renard for his valuable help and review of the uncertainty part. Thanks to the Plan Loire and Stephane Rodrigues from Tours University for lending us the US-P06 sampler.

## Statements and Declarations

Funding for the work was provided by the Compagnie Nationale du Rhône (CNR), Electricité de France (EDF), European Regional Development Fund (ERDF), Agence de l'eau RMC, and three regional councils (Auvergne-Rhône-Alpes, PACA and Occitanie) in the context of the Rhône Sediment Observatory (OSR, http://www.graie.org/osr). Any use of trade, firm, or product names is for descriptive purposes only and does not imply endorsement by the US Government. The authors declare no conflict of interest.

# Notations

## Roman symbols

| | |
|---|---|
| $C$ | Suspended sediment concentration $(\mathrm{g\,l^{-1}})$ |
| $\overline{C_{\mathrm{ISO}}}$ | Mean cross-sectional sand concentration calculated using the ISO method $(\mathrm{g\,l^{-1}})$ |
| $C_R$ | Bottom reference sediment concentration $(\mathrm{g\,l^{-1}})$ |
| $\overline{C_{\mathrm{rep}}}$ | Mean concentration per repetition set $(\mathrm{g\,l^{-1}})$ |
| $d$ | Depth of the vertical on the edge (m) |
| $D_*$ | Sedimentological diameter (-) |
| $\overline{C_{\mathrm{SDC}}}$ | Mean cross-sectional sand concentration calculated using the SDC method $(\mathrm{g\,l^{-1}})$ |
| $\overline{D_{50}}$ | Median diameter of the grain size distribution $(\mathrm{m^{-1}})$ |
| $g$ | Acceleration due to gravity $(\mathrm{m\,s^{-2}})$ |
| $i$ | Vertical MAP cell coordinate (-) |
| $j$ | Lateral MAP cell coordinate (-) |
| $h$ | Water depth (m) |
| $h_{\mathrm{cell}}$ | Cell height (m) |
| $k$ | Coverage factor (-) |
| $k_1$ | Coefficient relating the sand flux with the velocity Colby (1964) (-) |
| $l$ | Increment, sampling vertical (-) |
| $L_{\mathrm{edge}}$ | Length of the edge in ADCP measurements (m) |
| $\ln(C_{n_0}(z))$ | MaxPost best fitting vertical concentration profile $(\mathrm{g\,l^{-1}})$ |
| $\ln(C_{R,n_0})$ | MaxPost parameter $(\ln(\mathrm{gl^{-1}})$ |
| $m_{\mathrm{edge}}$ | Edge-shape exponent in ADCP measurements (-) |
| $m_{\mathrm{extrap}}$ | QRevInt extrapolation exponent (-) |
| $n_0$ | Parameters $\alpha$ and $\ln(C_R)$ used to calculate the $\ln(C_{n_0}(z))$ MaxPost profile (-) |
| $N_{\mathrm{rep}}$ | Number of repetitions per set (-) |
| $N_{\mathrm{sam}}$ Number of samples per vertical (-) | |
| $N_{\mathrm{seg}}$ | Number of segments (-) |
| $m$ | Number of verticals (-) |

| | |
|---|---|
| $u$ | Absolute standard uncertainty |
| $u'$ | Relative standard uncertainty (%) |
| $u'_\alpha$ | Uncertainty on the slope of the exponential profile $\alpha$ |
| $u'_{C_R}$ | Uncertainty on the reference concentration $C_R$ (%) |
| $u'_{\ln(C_R)}$ | Uncertainty on the logarithmic reference concentration $ln(C_R)$ (%) |
| $u'_h$ | Uncertainty in the elevation of the sampled point within the water column (%) |
| $u'_{lab}$ | Uncertainty due to laboratory analysis (%) |
| $u'_m$ | Random uncertainty due to the lateral integration (%) |
| $u'_{meas}$ | Uncertainty in the point concentration (%) |
| $u'_{nf}$ | Random uncertainty due to natural fluctuations in the suspended sediment concentration (%) |
| $u'_p$ | Total uncertainty due to the vertical integration (%) |
| $u'_{p,param}$ | Parametric uncertainty due to the vertical integration (%) |
| $u'_{p,struc}$ | Structural uncertainty due to the vertical integration (%) |
| $u'^2_Q$ | Uncertainty on the discharge of multiple transects (%) |
| $u'_{rep}$ | Relative standard deviation for each set of $N_{rep}$ repetitions (%) |
| $u'_{sampler}$ | Uncertainty due to the sampler type (%) |
| $\overline{u'_{sys,C}}$ | Uncertainty due to systematic errors in the suspended sediment concentration (%) |
| $u'_{sys,lab}$ | Uncertainty due to the systematic error of the laboratory analysis (%) |
| $u'_{sys,m}$ | Uncertainty due to the systematic error of the flux computation scheme (%) |
| $u'_{sys,p}$ | Uncertainty due to the systematic error of the vertical integration (%) |
| $u'_{sys,sampler}$ | Uncertainty due to the systematic error of the used sampler type (%) |
| $u'_{v*}$ | Uncertainty in the total shear velocity $v_*$ (%) |
| $u'_{ws}$ | Uncertainty in the settling velocity $w_s$ (%) |
| $u'_\kappa$ | Uncertainty in the von Karman constant $\kappa$ (%) |
| $u'_{\sigma t}$ | Uncertainty in the turbulent Schmidt number $\sigma$ (%) |
| $U$ | Absolute expanded uncertainty |
| $U'$ | Relative expanded uncertainty (%) |
| $U'_C$ | Relative expanded uncertainty in measurements of the suspended-sediment concentration (%) |
| $U'_m$ | Relative expanded uncertainty due to the lateral integration (%) |
| $U'_p$ | Relative expanded uncertainty due to the vertical integration (%) |

| | |
|---|---|
| $U'_\Phi$ | Relative expanded uncertainty in measurements of the suspended-sediment flux (%) |
| $U'_Q$ | Relative expanded uncertainty in measurements of the discharge (%) |
| $U'^2_Q$ | Relative expanded uncertainty on the discharge of multiple transects (%) |
| $U'_{\text{sys}}$ | Relative expanded uncertainty arising from systematic sources of error (%) |
| $p_s$ | Percentage of sand in the suspension (%) |
| $Q$ | Water discharge (m³ s⁻¹) |
| $q_{\text{ss}}$ | Suspended-sediment discharge per vertical (kg s⁻¹) |
| $s$ | Relative sediment density (-) |
| $v$ | Water velocity perpendicular to the cross-section (m s⁻¹) |
| $\overline{v}$ | Mean velocity (m s⁻¹) |
| $\overline{v_p}$ | Mean primary velocity (m s⁻¹) |
| $\overline{v_{0/l}}$ | Mean primary velocity from the closest measured vertical (m s⁻¹) |
| $v_*$ | Total shear velocity (m s⁻¹) |
| $w$ | Cell width (m) |
| $w_s$ | Settling velocity (m s⁻¹) |
| $x$ | Distance of the vertical from the start of the bank (m) |
| $X$ | East coordinate (MAP) |
| $X_{\text{proj}}$ | East coordinate on the average cross-section (MAP) |
| $Y$ North coordinates (MAP) | |
| $Y_{\text{proj}}$ | North coordinate on the average cross-section (MAP) |
| $y$ | Lateral coordinate |
| $y_{\text{lb}}$ | Left boundary of the cross-section (m) |
| $y_{\text{rb}}$ | Right boundary of the cross-section (m) |
| $z$ | Vertical coordinate |
| $z_a$ | Reference level for suspension at the top of the bedload layer (m) |
| $z_{\text{cell}}$ | Depth to the centreline of the cell (m) |

**Greek symbols**

| | |
|---|---|
| $\alpha$ | Vertical gradient of the exponential vertical concentration profile Camenen and Larson (2008). (-) |
| $\alpha_{R,n_0}$ | MaxPost parameter (-) |
| $\epsilon_v$ | Vertical diffusivity (m² s⁻¹) |

| | |
|---|---|
| $\epsilon_C$ | Relative difference in sand concentration determined by the SDC method compared to the ISO method (%) |
| $\epsilon_\Phi$ | Relative difference in sand flux determined by the SDC method compared to the ISO method (%) |
| $\theta$ | Shields parameter (-) |
| $\theta_{cr}$ | Critical Shields parameter or critical bed shear stress (-) |
| $\kappa$ | Von Karman constant (-) |
| $\mu$ | Parameter of the log-normal distribution |
| $\nu$ | Kinematic viscosity of water ($\mathrm{m^2\ s^{-1}}$) |
| $\xi$ | Ratio of the squared mean velocity to the total sampled depth Colby (1964) |
| $\sigma$ | Parameter of log-normal distribution, standard deviation of the normal distribution |
| $\sigma_t$ | Turbulent Schmidt number (-) |
| $\Phi_{\mathrm{ISO}}$ | Total suspended sand flux through a cross-section calculated using the ISO method ($\mathrm{kg\ s^{-1}}$) |
| $\Phi_{\mathrm{SDC}}$ | Total suspended sand flux through a cross-section calculated using the SDC method ($\mathrm{kg\ s^{-1}}$) |
| $\Phi_{\mathrm{total}}$ | Total suspended sand flux through a cross-section ($\mathrm{kg\ s^{-1}}$) |

*Competing interests.* TEXT

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
