# Peer review of "River suspended-sand flux computation with uncertainty estimation, using water samples and high-resolution ADCP measurements"

_EGUsphere, 2023_

## Referee Comment (RC2)

**Comments on "River suspended-sand flux computation with uncertainty estimation, using water samples and high-resolution ADCP measurements"**

**1 Summary**

This paper proposed a toolbox and a method to use high-resolution ADCP data and point-measured sand concentration data to estimate the suspended sand flux at a river cross-section. To achieve this, the paper proposes a method (SDC method) that employs the MAP method to interpolate the velocity field at the cross-section, then uses the BaM! method to estimate the vertical concentration profiles, and applies a physics-based approach for lateral interpolation. This work also includes uncertainty and error analysis of the SDC method. The toolbox is open-source and has been published online. The data used in this paper are also available for public access. This work performed a thorough analysis and quantified the error propagation in the sand flux measurement process; many of these errors are commonly neglected in other studies, which demonstrates the value and novelty of this work. How to use high-resolution measured data like ADCP data is always a question in scientific communities. The toolbox appears to provide a good solution to such an issue compare to the ISO method. This work is of good quality and worthy of publication. The main issue I encounter while reading this paper is the confusing definitions of different terms. This work focuses on the sediment transport theory proposed by Camenen and Larson (2008). There exists a variety of other classic sediment transport theories, which might have the potential to change the results of these analyses.

**2 Major comments**

1. Line 165. This work used an exponential function, or in other words, the linear profile, to describe the vertical profile of suspended sediment concentration. This is the result of assumption of constant vertical diffusivity (Line 164). However, it is known that the vertical diffusivity is not a constant in boundary layer flows. Is this linear form representative of the true profile? In Figure 5(b), one can observe a strong non-linear effect in the concentration data that is close to the bed. Is there a reason why the analysis was not performed using the Rouse profile or another similarly physically process-based profile to fit the vertical profile? Say

$$C(z) = C_R \left( \frac{(h-z)/z}{(h-z_a)/z_a} \right)^P \text{ where } P = \frac{w_s}{\sigma_t \kappa u_*}$$

2. There is a 2nd drawback using exponential fitting. In eqn. (1) quthor mentioned that the vertical profile is starting from a "reference level" $z_a$, if plug-in the $z_a$ into eqn. (4), have

$$C(z_a) = C_R \exp(\alpha z_a) \neq C_R \text{ unless } z_a = 0$$

This contradicts with the definition of $C_R$, that is the concentration at a reference level which is known not be 0 (Garcia, 2008; Wright and Parker, 2004; De Leeuw et al., 2020).

3. The definition of $\alpha$ needs a revise. In eqn.(4), eqn.(8) and line 201, $\alpha$ is negative. while in eqn.(5) and line 180 $\alpha$ is described positive. Here is an example. Line 180 states "$\alpha$ is strictly positive", while the definition of $\alpha$ in equation (8) denotes:

$$\alpha = -\frac{6w_s}{\sigma_t \kappa u_* h} < 0$$

4. Sec. 2.1.1 It is not clear that whether the Multitransect Averaged Profile (MAP) method is part of the contribution of this paper or not. The citation in this section pointing to an EGU talk. I did a quick google search and did not found detailed documentation on MAP method online. If this is part of the contribution of this paper, it will be nice to show more details on this method. Particularly, how the extrapolation to solid wall and to free surface are done (introduce the formulas etc.). If it is not part of this paper, please add a citation that introduced this method.

5. Line 172. Here briefly introduced BaM! method and mentioned that a large number (¿10 000) realizations are sampled. Please make it clear that these realizations are sampled by varying which parameter. In Line 176 says "BaM! simulator produces 500 realizations". Please state clearly what the number of realizations is in this study.

6. Line 207. The definition of $u'_\alpha$ is "relative uncertainty". While the $\sigma_\alpha$ is the standard deviation of $\alpha$, which should have the unit of "absolute standard uncertainty". It seems that the equation $\sigma_\alpha = u'_\alpha$ is not correct. Similar issue in Line 219.

7. Line 222. This is a recommendation. This study chose reference concentration formula that is proposed by Camenen and Larson (2008) to perform the analysis. There exists a variety of such formulas (see Garcia (2008) Chapter 2). I think the choice of reference concentration formula and the choice of formula for vertical suspended sediment concentration profile both could impact the uncertainty analysis. Has author considered the error caused by the choice of these theories/formulas?

8. Eq. (14). $\Phi_l$ is not defined. The last term has the form of flux weighted sum of $u'_{p,l}$. However, without clear definition of $\Phi_l$, one may suspect that $\Phi \neq \sum_l \Phi_l$

9. Line 416 and Figure 8. The result demonstrates that the uncertainties in all these river data are dominated by the vertical integration $U'_p$. It might worth to look at how much changing the sediment transport theory, such as reference concentration equation and vertical profile theory could impact the results.

**3  Other comments**

1. The standard cross-referencing of equations should have a parenthesis, e.g., Eq. (1). Figure caption should be bold etc. See format requirement from eSurf website: https://www.earth-surface-dynamics.net/submission.html#templates

2. The format of References also need to be revised.

3. The figure captions are not detailed enough. For example, in Figure 5 and 6, there is no explanation in the figure or in the caption to tell what the meaning of the different lines and the numbers in legends are.

4. Line 55. The VMT is mentioned several times in the paper without indicating its full form. I suggest to provide the full name of it in the introduction.

5. Similarly, for other abbreviations, e.g., please define "BaM!" method in Line 168. MaxPost profile in Line 176. QRevInt in Line 57 etc., it would be nice to give full definition of them.

6. Line 119-124. The choice of symbols $u, u', U, U'$ to represent different uncertainties is confusing. It is convention in fluid mechanics that $u$ and $U$ represent flow velocity; $u'$ and $U'$ represent velocity fluctuations. The author uses the velocity $u$ in Equation (1) as well. I recommend use other symbols to represent these uncertainty metrics.

7. Line 239. Another similar issue, $h$ is defined as water depth in Line 92, while it is used to describe cell height here. It is better to use another symbol for cell height.

8. Line 224, please define what is $h_j$.

9. Line 119-124. A second suggestion on the terminologies. From description, here is my understanding, $u$ is standard deviation $\sigma$ (which is pointed out in the text). $u'$ is the coefficient of variation $CV$. $U$ is similar to $k\sigma$ the z-score, etc. It will be helpful to include these common concepts in statistics in their definitions. My understanding might be correct or might be wrong. So please define clearly what are these terms (using equations) to avoid confusing.

10. Line 187. In Eqn.(6), the constant should be $1.5 \times 10^3$

**References**

De Leeuw, J., Lamb, M. P., Parker, G., Moodie, A. J., Haught, D., Venditti, J. G., and Nittrouer, J. A. (2020). Entrainment and suspension of sand and gravel. *Earth Surface Dynamics*, 8(2):485–504.

Garcia, M. (2008). Sedimentation engineering: processes, measurements, modeling, and practice. American Society of Civil Engineers.

Wright, S. and Parker, G. (2004). Flow resistance and suspended load in sand-bed rivers: simplified stratification model. *Journal of Hydraulic Engineering*, 130(8):796–805.

---

## Author Response (AR1)

**Comments Reviewer 1:**

The authors introduce a method to determine the sand flux in a river cross-section based on water samples and ADCP measurements, with a systematic uncertainty estimate. The team of authors have a long history in the topic, and build on much of their previous work. I am very positive about the fact that the authors aim to publish their work in peer-reviewed literature, because monitoring methods like this often remain described in grey literature. At the same time, my review below is very critical. Frankly, I believe there is too much uncertainty in the uncertainty estimation, and personally I would not adopt this approach. Despite this, once again, I appreciate the effort and I am convinced the approach deserves to be published after revision.

**Major points**

1. The velocity field is treated completely independent from the suspended sand concentration (SSC) field, while the latter depends on the former. Standard SSC profiles such as the Rouse profile proceed from crude assumptions such as a steady state, and neglect of streamwise and lateral SSC gradients. These assumptions are never truly met, and it would be better to use the measured velocity profiles to derive a theoretical SSC profile, or to use the acoustic backscatter profiles. The authors acknowledge in their discussion that use of ADCP backscatter would be a promising way forward. They argue that the use of ADCP backscatter for SSC monitoring is also prone to uncertainty. This is true for calibration methods with the ultimate aim to translate ADCP backscatter profiles to SSC without direct SSC samples in the profile, but not so much if the aim is to extrapolate. My point is that if one has an ADCP backscatter profile and three samples at different depths across that same profile, and SSC is to be extrapolated over the complete vertical, then there really is no need to rely on a theoretical profile based on assumptions of a steady flow and no horizontal gradients.

A: Thank you for your comment, we agree that the sand concentration field depends on the flow velocity field, for example when pronounced lateral gradients exist. In our study, both measurements are performed independently, using ADCP measurements for the velocity field and sampling for the concentration field. Nevertheless, we used the ADCP measurements of the velocity field for the vertical integration of the concentration using Bayesian modeling. To do so, we determine the mean vertical velocity $\overline{v}_l$ at each sampling vertical $l$, which is used to estimate the priors on $\alpha$ and $C_R$ (Equations 6 and 8) based on semi-empirical equations by Camenen & Larson (2008). The mean vertical velocity $\overline{v}_l$ is used to calculate the Shields parameter $\theta$ required for the determination of the reference concentration $C_R$ (Equation 6):

$$\theta = \frac{0.5 f_c \overline{v}_l^2}{(\rho - 1)\overline{d_{50,l}}g}$$

where $f_c$ is the skin friction coefficient and $\overline{d_{50,l}}$ the mean vertical median grain size. It is also used to calculate the total shear velocity $v_*$, required to determine $\alpha$ (Equation 8):

$$v_* = \overline{v}_l \sqrt{\frac{f_{c,v*}}{2}}$$

where $f_{c,v*}$ is the total friction coefficient (including effects of bedforms) for the total shear velocity. In the proposed SDC method, we implemented the Camenen & Larson formula, because it does not assume a zero concentration at the surface as the Rouse profile. However, simple empirical fits using the exponential Camenen & Larson and the Rouse profile are included in the toolbox as well (see below for a detailed discussion on these profiles).

We noted the use of ADCP measurements for the vertical integration in section 2.2.3, e.g.: *"The expected values of α and $C_R$ are evaluated based on local hydro-sedimentary parameters (Camenen and Larson, 2008), which are determined using ADCP depth and velocity, and bedload measurements."*

Concerning the use of standard SSC profiles or ADCP measurements for the vertical integration, we followed the standard approach and a standard (exponential) SSC profile to fit the data. We agree, as noted in our discussion, that using the ADCP backscatter is promising and would move forward the interpolation of the concentration in the cross section. The major advantage is the higher spatial resolution of the ADCP measurements compared to the sampling. In contrast, the main issue with the backscatter profiles we observed is that they are very noisy, because they are not time averaged. Using the backscatter is an important perspective and should be implemented soon into the proposed method. There are however still some issues in the acoustic processing to obtain smooth profiles. We clarified the point in the discussion:

*"A major advance in both the vertical, but particularly the lateral integration, may be made using the acoustic backscatter measured by the ADCP. One option is the use of the backscattered signal intensity as a proxy for suspended-sediment concentration facilitating the inter- and extrapolation in between the sampling points. Moreover, several studies…"*

2. In their uncertainty analysis, the authors assume that errors in Q and C are independent, which I think is unlikely. The rivers subject to monitoring feature transient secondary flow cells, or coherent flow structures, causing that deviations from the dominant flow and SSC fields go hand in hand. Could the authors prove from their data that indeed errors in Q and C are mutually independent?

A: Thank you for your comment, we clarified our approach in the text. The proposed SDC method focuses on applications in rivers at simple sampling sites without tidal effects or strong secondary currents. If secondary flow cells or coherent flow structures cause deviations in SSC from the dominant flow equilibrium, such deviations should be evaluated independently if sufficient measurement verticals are provided. If not, it would increase uncertainties since our methodology assumes a dominant flow equilibrium for the lateral interpolation of concentrations. Choosing an appropriate measurement site is thus primordial to allow for correct suspended sediment ratings. We detailed the characteristics of study site. Changed to:

*"The SDC method focuses on suspended sand flux measurements in simple river cross-sections without tidal effects or strong secondary currents. In case of strong secondary flow cells causing deviations in the suspended sand concentration from the dominant flow equilibrium, these deviations should be evaluated separately. Choosing an appropriate measurement site is essential to obtain reliable sand flux and uncertainty estimates (Edwards and Glysson, 1999)."*

Although the concentration field depends on the velocity field, we suppose that the errors in discharge and concentration can be assumed as independent. First, the velocity and concentration measurement methods are almost independent; ADCP measurements are used for the velocity, and suspended sediment samplers and filtration to estimate the concentration. We assume that using the ADCP velocities and depths for the interpolation of the concentration and the estimation of the priors is only a limited contribution. As soon as the ADCP backscatter is used for the interpolation of the concentration, the measurement methods are not independent anymore; then, the influence on the errors in discharge and concentration has to be evaluated and maybe the calculation adjusted.

However, velocities are evaluated from the Doppler effect while concentrations are correlated through the backscattering field, which could be assumed as independent.

Second, the error sources in discharge and concentration are significantly different. The errors in discharge originate notably from the extrapolation for the velocities in the unmeasured areas close to the surface, the river bottom or the banks. In contrast, the errors in the concentration originate mostly from the discretization of the cross section, which we divided into the uncertainties in the lateral and vertical integration of the concentration. This error is magnified by the limited resolution of the samplings, whereas the resolution of the ADCP velocity measurements is much better and the related error can be assumed to be negligible. Other substantial errors in concentration are the laboratory measurement or the sampler type, which are both independent from the discharge.

We included a short paragraph on the independence of the errors in discharge and concentration as well as our assumptions in section 2.3.1:

*"Equation (15) is based on the hypothesis that the errors in discharge and concentration are independent, otherwise, the term would have to include the associated covariances. Such assumption that the errors are independent appears however reasonable. First, because discharge and concentration are measured independently, the discharge being measured using ADCPs and the concentration being determined using sampling and laboratory analyses. Second, the error sources in discharge and concentration are significantly different. Velocity lateral interpolation errors are negligible due to the high spatial resolution of ADCP measurements whereas concentration lateral interpolation errors are large. We agree that these two error components are physically correlated, but this is not a problem as the first one is negligible."*

And in the discussion (section 4.1.):

*"However, when using the ADCP backscatter for the calculation of the concentrations, the errors in discharge and concentration would probably not be independent anymore, because the same measurement method is used and the uncertainty due to the spatial integration of the concentration, the major error source of the uncertainty in concentration, will decrease. Consequently, the formulation of the uncertainty estimation should be adapted, e.g. by including the covariances."*

3. The previous comment addresses just one out of a long list of assumptions that had to be made in the uncertainty analysis. How can the quality of the uncertainty analysis be evaluated? In the end, I would be more comfortable with an approach in which an extremely large dataset is used to investigate the loss of accuracy when fewer field samples are available. The authors claim that their method reduces the time and effort required for sampling, compared to ISO standards. This comes at a cost though, and this cost remains unclear to me. Wouldn't it make more sense to take a dataset that fully complies with the ISO standards, and then show how the new method yields similar results with less data?

A: Obtaining a data set complying with the ISO standard requires a huge number of samples, not only to estimate the concentration, but notably to estimate the uncertainty in the concentration. For a 100 m wide river, at least seven verticals and five points are required to estimate the concentration, which yields 35 samples in total. For the uncertainty estimation, all error sources are estimated by separated repeated sampling measurements. To estimate the uncertainty in the vertical integration only, a total of 420 samples is recommended, divided into more than 30 verticals with seven points per vertical and sampling every point twice. Due to this considerable number of samples required, it would be extremely difficult to prepare a dataset following the ISO recommendations. In most natural rivers, the

conditions are changing too fast to collect such a high sample number under steady conditions. It would be necessary to perform these experiments under different hydro-sedimentary conditions in the laboratory. This contradicts the applicability of the ISO uncertainty estimation method for monitoring purposes in the field.

We have been searching for different data sets following the ISO approach, however, we have not found any that fully applies it.

4. The authors fully focus on sand concentrations, whereas suspended sediment concentration estimates from samples typically include the fine sediment part. The authors tend to neglect the complexity of fine sediment dynamics. On line 25 they write "fine suspended sediments (…) are relatively homogeneous throughout the cross section". On line 425: "However, as only sand concentrations are considered here, the percentage of sand was set equal to 100%." That seems crude. The spatiotemporal variation in fine sediment concentrations needs a more nuanced treatment and deserves a place in the uncertainty analysis.

A: In this article, we are focusing on the measurement of suspended sand and its associated uncertainty. Compared to suspended fine sediments, suspended sands show much larger spatio-temporal gradients and measurement techniques are less advanced and comparable. Moreover, suspended sand monitoring is rarely performed in many countries across the world, whereas suspended fine sediment monitoring is more popular. Similar to the procedure for acoustic suspended sediment measurements, grain size analysis or concentration measurements, we separate the fine sediment from the sand fraction.

We reformulated as: "*Even though suspended-sand transport is a key driver of the river evolution (Kondolf, 1997), it remains difficult to measure its concentration due to its temporal and spatial variability in the cross-section which may account for several orders of magnitude (Armijos et al., 2017). In contrast, fine suspended sediments (<63 μm) are relatively homogeneous throughout the cross-section, often reaching concentration variations of up to an order of magnitude (Wren et al., 2000).*"

Using 100 % as the percentage of sand to estimate the uncertainty due to the lateral integration is recommended by the authors of the used nomograph. This does not necessarily signify that no fine sediments are present in the suspension; but refers only to this part of the uncertainty estimation. Supposing 100 % increases the uncertainty due to the lateral integration, compared to lower percentages. We detailed this choice in the related section:

"*If only the suspended-sand flux through a cross-section is measured or is of principal interest, as in our study, the percentage of sand in the equation should be supposed to be 100 %, neglecting the influence of the fine-sediment flux. This does not mean that no suspended fine sediments are present, but this allows to account for the considerable lateral gradients of the sand suspension. It also avoids underestimating the uncertainty due to the lateral integration, because the uncertainty $u'_m$ for the same sediment discharge measurement (same m and ξ) is higher, when assuming $p_s = 1$ than including fine sediments.*"

**Line by line comments**

66 "However, acoustic inversion techniques require many physical samples for calibration, and are affected by acoustic modelling issues (Vergne et al., 2023)." See major point 1.
A: Paragraph changed to: *"Moreover, the acoustic backscatter measured by an ADCP, or an Acoustic Backscattering System (ABS) may be used to improve the spatial integration of the concentration in the cross-section. Indeed, the acoustic backscatter can be inverted and used to measure the suspended-sand concentration (e.g. Topping and Wright 2016; Venditti et al. 2016; Szupiany et al. 2019; Vergne et al. 2020). Several software tools have been developed to process ADCP data for estimating suspended-sand flux (Boldt, 2015; Dominguez Ruben et al., 2020) or using backscatter inversions. However, acoustic inversion techniques require many physical samples for calibration, and are affected by acoustic modelling issues (Vergne et al., 2023)."*

78-81 please better explain
A: Corrected to: *"It identifies several sources of error of random and systematic nature. These errors are notably related to the lateral integration of the concentration in between the sampling verticals, thus depending on the number of verticals. Another important error source is the vertical integration of the concentration in between the sampling points of a vertical, which depends on the number of sampling points along a vertical. Another error is related to the sampling time and consists of the sample's representativeness of the natural fluctuations of the concentration due to turbulence. Additional error sources originate from the sampler type and the laboratory analysis. The uncertainty related to each of these error sources is estimated by performing a high number of samples, whereof the average is considered as the approximate true value. This value is taken as reference and the uncertainties originating from the different error sources are estimated based on the deviation from the reference."*

85 "the large amount of additional samples required for the uncertainty analysis is not realistic to apply" Not in Europe, but this is different in China. Please acknowledge this.
A: We are not aware of any Chinese data set which may be used for the uncertainty analysis. We changed this sentence to highlight the importance of stable hydro-sedimentary conditions: *"Second, the large number of additional samples required for the uncertainty analysis and taken under stable hydro-sedimentary conditions is hardly applicable in most environments due to the quick variability of the hydro-sedimentary processes and technical difficulties."*

90 Please better motivate the approach of Colby (1964), explaining when it is applicable.
A: Changed to: *"Concerning lateral integration, Colby (1964) noticed that the sand flux varies approximately (within a factor $k_1$) with the third power of the mean velocity $\overline{v}$ for a constant grain size distribution and temperature, and velocities ranging between 0.6 and 1.5 m/s: $\Phi_{total} = k_1\overline{v}^3$. Based on these observations and making the required conversions, he stated that the variability of sand concentration at different sampling verticals should be closely related to the variability of $\overline{v}^2/h$, the ratio of the squared mean velocity $\overline{v}$ to the total sampled depth h."*

95 you forgot h_l
A: Corrected, thank you.

139-140 How does the QRevInt optimized extrapolation law compare with the approach by Vermeulen et al. (2014)?
A: It seems to us that Vermeulen et al. (2014) proposed a method to establish the mean velocity field that is different from the method applied by QRevInt. They aim to combine the measurements spatially close to each other, notably far from the ADCP, where the distance between the beams is high. In contrast, QRevInt proceeds a temporal mean, so that cells of the same ADCP ensemble are averaged. Included in the introduction: *"Other examples are QRev*

*(Mueller, 2016) and QRevInt (Lennermark and Hauet, 2022), which are applied to ensure discharge measurement reliability and to quantify the uncertainty in the discharge measurement (Despax et al., 2023). While these methods provide temporal averages, that is of each ADCP ensemble, other approaches combine the ADCP measurements spatially close to each other, notably far from the instrument, where the distance between the beams is high (Vermeulen et al., 2014)."*

Equation 8: where does the 6 in the equation come from? I would appreciate a derivation (just for me to check).
A: It comes from the vertical integration of the Rouse profile (cf. Camenen & Larson, 2007, 2008). To be homogeneous for the Schmidt number using the exponential profile or the Rouse profile, Camenen & Larson (2008) assumed the Rouse profile as a reference and calculated the other profiles based on it., i.e. having the same depth-averaged diffusion coefficient for all profiles.

210 How are those uncertainty numbers verified? There is large uncertainty in these uncertainty estimates.
A: Only few studies evaluated these uncertainty numbers. We used the values as found in the literature without performing additional laboratory or other experimental measurements, which are beyond the scope of this article. Clarified to: *"Only a few studies evaluated these uncertainties. Since additional experimental measurements are beyond the scope of this article, we define the uncertainty based on literature, ..."*

275 Ta facilitate -> To facilitate
A: Corrected, thank you.

340 "In the best case, these measurements should be conducted on every sampling campaign, however, in reality, this is not possible." It is costly, but possible, and worth doing.
A: Based on our suspended sand monitoring and sampling experience, a suspended sand sampling takes two to five hours including 20 to 30 samples taken using the US P-6 sampler on a cableway or the Delft bottle or using a truck. Considering the variability of the discharge and thus the sand concentration, this is a very long sampling time. It should not be increased to avoid sampling under changing conditions. Using this same protocol, we were not able to significantly decrease the sampling time, e.g. due to the displacement velocity of the cableway or the sampling duration required for Delft bottle samples (at least five minutes) and the time required to clean the sampler in between two samples. In contrast, we have made good experiences with a pump sampler which yields results consistent with the US P-6 sampler and can be deployed much faster. Additionally, it is also possible to increase the number of samplers so that several samples can be taken simultaneously. Being able to perform samples faster allows to take more samples for the uncertainty estimation as well. However, we believe that the number of samples required for the uncertainty estimation should be in a reasonable ratio to the number for the concentration measurement, e.g. the number of samples required for the uncertainty estimation should not exceed the number for the concentration measurement.

Changed to: *"In the best case, these measurements should be conducted on every sampling campaign, however, in reality, this is hardly possible using the presented measurement techniques. Using the same sampling protocol, the sampling campaign with additional samples for the uncertainty estimation would take very long, so that the variation in river discharge and concentration would become too large."*

341 "The sampling campaign with additional samples for the uncertainty estimation would take very long, so that the variation in river discharge would become too great." One could collect samples at a higher frequency.
A: See response and changes above.

385 Does the behavior becomes less Rousean with flow strength, or size of the river?

No, the Rouse model should be dimensionless (as well as the exponential profile model). The diffusion coefficient is inversely proportional to the water depth (Eq. 10), which could be one explanation. However, based on Eqs. (7) and (10), one obtains $C(h)/C(0) = \exp[-6w_s/\sigma_t \kappa u_*]$, which is mainly a function of the shear velocity (and grain size for the settling velocity). The presence of dunes, whose height is typically a function of the water depth can affect the value of the shear velocity. Nevertheless, as noticed by Van Rijn (1984) or Santini et al. (2019), the Rouse model is not correct close to the surface as it yields zero-concentration while positive values have often been observed.

Changed to: *"(cf. Fig. 6d). This indicates a possible effect of the water depth on the concentration gradient through the vertical diffusion coefficient (inversely proportional to the water depth, Eq. (10)) and/or due to dunes (whose height is typically proportional to the water depth, and so can enhance the shear velocity)."*

465 Please differentiate between inter- and extrapolation, and calibration to measure stand alone.

A: Changed to: *"Nevertheless, for single-frequency applications as the ADCP, strong assumptions or calibrations are necessary to invert correctly the concentration and grain size of silt-clay and sand-sized sediment (Vergne et al., 2023)."*

480 This is what Vermeulen et al. (2014) do.

A: We added this citation, thank you.

495 "differences from the results from the ISO method are up to 15%" In fig. 10 I see differences exceeding 35%. I assume you claim your method is an improvement. How can you quantify/prove this?

A: We claim that our proposed method is more flexible to different measurement protocols and more easily applicable, in terms of the concentration and uncertainty estimation methods. In contrast, we do not claim that the SDC method is an improvement compared to the ISO method in terms of the results. Since no references are available, the best method cannot be determined. Extensive field or laboratory measurements under stable and known conditions are required to assess the performance of these two and other existing methods.

We corrected the value and clarified to: *"The relative differences between the sand fluxes calculated following the SDC method compared to the ISO method may reach up to 40 %, but no bias is visible. Extensive field and laboratory measurements under stable and known conditions are required to evaluate the performance of the two methods and determine the best among these two and other existing methods. Divergences may originate from the difference in the sampling protocols. The ISO method is based on a sampling protocol with seven points per vertical, where the samples are taken at precise relative depths, while four samples are typically taken per vertical and at varying relative depths in the present measurements."*

506 Compared to the ISO method, this reduces considerably the required sampling time and effort. Yes, but at the cost of a large number of assumptions.

A: The proposed uncertainty estimation method is a first proposition that still requires many improvements and advances at the long-term scale to include all error sources and estimate them precisely. Similar to the long development required to estimate the uncertainty in ADCP discharge and velocity measurements, this first framework may serve as a basis for future developments. As far as the authors know, no study applied the ISO uncertainty method, which questions its popularity and applicability. Despite many assumptions, we are convinced that our proposed method responds to scientific and operational needs of estimating the uncertainty in suspended sand concentration measurements with a reasonable time and effort required.

We changed to: *"Compared to the ISO method, this reduces considerably the required sampling time and effort, facilitating its application even for monitoring purposes. The price for this is additional*

*assumptions on the spatial distribution of concentrations, based on simple physical basis valid for gauging cross-sections with a simple flow."*

515 Kästner et al. (2018) offers an in-depth analysis of lateral velocity profiles, which may help to put this in a broader context.
A: Reference added, and sentence changed to: *"Developing a consistent method for both uses and adapted to various channel geometries and lateral concentration gradients is needed, e.g. following approaches developed for index-velocity relations (e.g. Kästner et al. (2018))."*

Kästner, K., Hoitink, A. J. F., Torfs, P. J. J. F., Vermeulen, B., Ningsih, N. S., & Pramulya, M. (2018). Prerequisites for accurate monitoring of river discharge based on fixed-location velocity measurements. Water Resources Research, 54(2), 1058-1076.

Vermeulen, B., Sassi, M. G., & Hoitink, A. J. F. (2014). Improved flow velocity estimates from moving-boat ADCP measurements. Water Resources Research, 50(5), 4186-4196.

**Comments Reviewer 2:**

This paper proposed a toolbox and a method to use high-resolution ADCP data and point-measured sand concentration data to estimate the suspended sand flux at a river cross-section. To achieve this, the paper proposes a method (SDC method) that employs the MAP method to interpolate the velocity field at the cross-section, then uses the BaM! method to estimate the vertical concentration profiles, and applies a physics-based approach for lateral interpolation. This work also includes uncertainty and error analysis of the SDC method. The toolbox is open-source and has been published online. The data used in this paper are also available for public access. This work performed a thorough analysis and quantified the error propagation in the sand flux measurement process; many of these errors are commonly neglected in other studies, which demonstrates the value and novelty of this work. How to use high-resolution measured data like ADCP data is always a question in scientific communities. The toolbox appears to provide a good solution to such an issue compare to the ISO method. This work is of good quality and worthy of publication. The main issue I encounter while reading this paper is the confusing definitions of different terms. This work focuses on the sediment transport theory proposed by Camenen and Larson (2008). There exists a variety of other classic sediment transport theories, which might have the potential to change the results of these analyses. Please find the detailed comments in the PDF attached.

A: Thank you for your review and comments. We hope that the corrections and changes clarify the text.

Concerning your remark about the notations: We used "u" for the uncertainties as commonly used in metrology, and "v" for the velocities to avoid confusion and because only the total shear velocity $v_*$ is used (see also our response to your other comments 6.). We have checked again the manuscript and provided a list of notations in the appendix for convenience.

Concerning the used vertical concentration profile: We tested empirically fitted Rouse and Camenen and Larson profiles prior to the implementation into the SDC method. The provided toolbox includes in addition to the presented SDC method a similar option using the Rouse vertical profile. In this socalled SDC-Rouse method, we fitted the Rouse profile directly to the sampling points instead of fitting the exponential profile using Bayesian modelling as for the SDC method. Except this more traditional and widely used approach for the vertical integration, the lateral integration and determination of the point concentration remain the same.

The relative differences between the three different methods (ISO, SDC, SDC-Rouse) are usually smaller than the estimated uncertainty in the flux $U'_\Phi$. Figures like Figure 11 in the main part of the article are included in the supplementary material, comparing the SDC-Rouse with the ISO and SDC methods, respectively. Except the relative difference $\varepsilon_{\Phi,Rouse-ISO} = (\Phi_{Rouse} - \Phi_{ISO})/\Phi_{ISO}$ calculated for the Amazon, which accounts for 32 %, is larger than the uncertainty $U'_\Phi$ (26.6 %). This shows that the uncertainty in the flux measurement is larger than the difference between the applied methods, but also that many error sources exist. Consequently, the differences between the used vertical concentration profiles are considered as negligeable.

At the same time, these results show the importance of the vertical integration and its contribution to the result. In the SDC method (either using the exponential or Rouse profile), the parameters of the vertical concentration profiles are used for the lateral integration, thereby increasing the importance of well fitted profiles. Finally, these comparisons highlight the importance of the number and position of the sampling points along the vertical and the possible improvement of the spatial integration using high-resolution hydro-acoustic measurements of the cross-section.

Based on these results, we decided to implement the exponential profile (as proposed by Camenen and Larson, 2008) into the SDC method using Bayesian modeling to fit the profile. In addition to this exponential profile, others may be included as well. An advantage of the Camenen and Larson profile is that it does not require the definition of a reference level, simplifying the fit and Bayesian modeling. In contrast, the Rouse profile assumes a zero-concentration at the surface and requires the introduction of a reference level close to the bed.

In section 2.2.3 we added: *"Prior to the implementation of the Camenen and Larson (2008) profile, we compared empirically fitted Camenen and Larson (2008) and Rouse (1937) profiles. The differences in sand fluxes are lower than the uncertainty in sand flux of the respective measurement (see supplementary material). Moreover, the profile of Camenen and Larson (2008) has a practical interest that it does not request to arbitrarily define a reference level. Consequently, we used only the Camenen and Larson (2008) profile for the Bayesian modeling in the SDC method. However, other vertical concentration profiles may be used as well and included into the toolbox."*

In section 4.1. we included: *"Moreover, the use of the parameters of the vertical concentration profile for the lateral integration increases the importance of a well-fitted vertical profile, otherwise, large uncertainties may result."*

In section 4.3. we included: *"The relative difference $\varepsilon_\Phi$ ranges for all four studied rivers between -40 % and +23 % with no clear relationship with the total flux (Fig. 11). These values are lower than the estimated uncertainties, underlining the importance of the errors associated to the sand flux measurement. Similar observations can be made when using an empirically fitted Rouse profile instead of the exponential profile fitted with Bayesian modeling (see supplementary material). In comparison..."*

**Major comments**
1. Line 165. This work used an exponential function, or in other words, the linear profile, to describe the vertical profile of suspended sediment concentration. This is the result of assumption of constant vertical diffusivity (Line 164). However, it is known that the vertical diffusivity is not a constant in boundary layer flows. Is this linear form representative of the true profile? In Figure 5(b), one can observe a strong non-linear effect in the concentration data that is close to the bed. Is there a reason

why the analysis was not performed using the Rouse profile or another similarly physically process-based profile to fit the vertical profile? Say

$$C(z) = C_R \left( \frac{(h-z)/z}{(h-z_a)/z_a} \right)^P \text{ where } P = \frac{w_s}{\sigma_t \kappa u_*}$$

*As discussed by Camenen and Larson (2007), while fitting an exponential or a Rouse-type profile to experimental data, one obtains very similar results having in mind the uncertainties of the experimental data. Even if the Rouse profile should be more accurate close to the bed, it also deviates from reality very close to the bed, where it assumes infinite concentrations for z close to zero. The exponential profile has a practical interest that it does not request to arbitrarily define a reference level. We compared the results obtained using the Rouse and exponential profile and included them into the supplementary material.*

*We included the following paragraph in section 2.2.3: "Prior to the implementation of the Camenen and Larson (2008) profile, we compared empirically fitted Camenen and Larson (2008) and Rouse (1937) profiles. The differences in sand fluxes are lower than the uncertainty in sand flux of the respective measurement (see supplementary material). Moreover, the profile of Camenen and Larson (2008) has a practical interest that it does not request to arbitrarily define a reference level. Consequently, we used only the Camenen and Larson (2008) profile for the Bayesian modeling in the SDC method. However, other vertical concentration profiles may be used as well and included into the toolbox."*

2. There is a 2nd drawback using exponential fitting. In eqn. (1) author mentioned that the vertical profile is starting from a "reference level" za, if plug-in the za into eqn. (4), have

$$C(z_a) = C_R \exp(\alpha z_a) \neq C_R \text{ unless}$$

This contradicts with the definition of $C_R$, that is the concentration at a reference level which is known not be 0 (Garcia, 2008; Wright and Parker, 2004; De Leeuw et al., 2020).

*A: The interest of the exponential profile is that it does not necessitate to define a reference level. The concentration tends to $C_R$ if z tends towards 0 (i.e. $C(z_a) \approx C_R$ if $z_a$ close to zero). Indeed, for most other models (Rouse profile), the concentration tends towards infinite if z tends towards 0, which necessitates to define a reference level $z_a > 0$. The reference concentration for the exponential profile is clearly different from the reference concentration for a Rouse profile. For the latest, the reference concentration is sensitive to the (more or less arbitrary) choice of the reference level, which adds some complexity to evaluate the priors. We included into section 2.2.3: "This reference concentration $C_R$ differs significantly from the reference concentration for a Rouse profile, where the reference concentration is sensitive to the (more or less arbitrary) choice of the reference level adding some complexity to evaluate the priors."*

3. The definition of $C_R$ needs a revise. In eqn. (4), eqn. (8) and line 201, α is negative. while in eqn. (5) and line 180 α is described positive. Here is an example. Line 180 states "α is strictly positive", while the definition of α in equation (8) denotes:

$$\alpha = -\frac{6w_s}{\sigma_t \kappa u_* h} < 0$$

*A: We homogenized the notation and definition of $\alpha$ throughout the text, thank you. In equation 7 (former eq. 5), we now use the absolute value of α, to ensure increasing concentrations with increasing depth.*

4. Sec. 2.1.1 It is not clear that whether the Multitransect Averaged Profile (MAP) method is part of the contribution of this paper or not. The citation in this section pointing to an EGU talk. I did a quick

google search and did not find detailed documentation on MAP method online. If this is part of the contribution of this paper, it will be nice to show more details on this method. Particularly, how the extrapolation to solid wall and to free surface are done (introduce the formulas etc.). If it is not part of this paper, please add a citation that introduced this method.

A: Thank you for the remark. MAP has only been published at the EGU and then implemented into QRevInt. We have included a precise description in the supplementary materials and detailed the method more in section 2.2.1.

5. Line 172. Here briefly introduced BaM! method and mentioned that a large number (>10 000) realizations are sampled. Please make it clear that these realizations are sampled by varying which parameter. In Line 176 says "BaM! simulator produces 500 realizations". Please state clearly what the number of realizations is in this study.

A: The BaM! method performs 10,000 realizations; they are all included to determine the best fitting (maxpost) profile. Of these 10,000 realizations, only 500 are registered and can thus be reused for the further uncertainty analysis.

Changed to: "*The first 5,000 realizations are burnt as a warm-up period, then the last 5000 realizations are decimated to decrease the correlation in the result.*" And later ". *The posterior distribution of the parameters is computed using Bayes theorem, and a large number (> 10,000) of realizations are sampled using an adaptive block Metropolis Markov Chain Monte Carlo (MCMC) sampler (Renard et al., 2006) varying the parameters α and ln(C(z))."*

6. Line 207. The definition of $u'_\alpha$ is "relative uncertainty". While the $\sigma_\alpha$ is the standard deviation of α, which should have the unit of "absolute standard uncertainty". It seems that the equation $\sigma_\alpha = u'_\alpha$ is not correct. Similar issue in Line 219.

A: We use this approximation for lognormal distributions, assuming that its σ-parameter is approximately the covariance. The covariance $cov$ of a lognormal distribution is:
$$cov = \sqrt{\exp(\sigma^2) - 1}$$
Performing a series expansion to 0 ($x \to 0$), yields σ:
$$\exp(x) = 1 + x + o(x)$$
$$\exp(\sigma^2) - 1 = \sigma^2 + o(\sigma^2)$$
This approximation may be made for σ < 0.5.

7. Line 222. This is a recommendation. This study chose reference concentration formula that is proposed by Camenen and Larson (2008) to perform the analysis. There exists a variety of such formulas (see Garcia (2008) Chapter 2). I think the choice of reference concentration formula and the choice of formula for vertical suspended sediment concentration profile both could impact the uncertainty analysis. Has author considered the error caused by the choice of these theories/formulas?

A: We have not evaluated the error related specifically to the use of the exponential profile compared to another profile (and other semi-empirical formulas associated for the reference concentration). The used exponential profile is one option among many possible concentration profiles. As indicated above, we compared using an empirically fitted Rouse profile with the SDC method. Their differences are smaller than the estimated uncertainties. Depending on the measurement and the hydro-sedimentary conditions, another formula than the Camenen and Larson (2008) may be the most appropriate. For a Rouse profile, one should use a specific semi-empirical formula for the reference concentration as an a priori such as the McLean (1992) formula. As a perspective, other formulas may be included, and the best fitting formula may be chosen limiting thereby the uncertainty in the vertical integration.

We included into section 4.4: "*Other vertical concentration profiles may be included, and the best fitting formula may be chosen limiting thereby the uncertainty in the vertical integration.*"

8. Eq. (14). $\Phi_l$ is not defined. The last term has the form of flux weighted sum of u0 p;l. However, without clear definition of $\Phi_l$, one may suspect that $\Phi \neq \sum_l \Phi_l$

A: We included the following definition: "*and $\Phi_l$ is the suspended sediment flux through vertical l."*

9. Line 416 and Figure 8. The result demonstrates that the uncertainties in all these river data are dominated by the vertical integration $U'_p$. It might worth to look at how much changing the sediment transport theory, such as reference concentration equation and vertical profile theory could impact the results.

A: As noted above, we have not separately investigated the error related to the used vertical concentration profile. The performed comparisons with the Rouse profile, however, showed limited difference between the two profiles. Moreover, in the uncertainty due to the vertical integration $U'_p$, the uncertainties in the point concentration are included as well. Those are the uncertainty due to the laboratory analysis, the natural fluctuation and the sampler type.

**Other comments**

1. The standard cross-referencing of equations should have a parenthesis, e.g., Eq. (1). Figure caption should be bold etc. See format requirement from eSurf website: https://www.earth-surface-dynamics.net/submission.html#templates

A: Corrected, thank you.

2. The format of References also need to be revised.

A: Corrected, thank you.

3. The figure captions are not detailed enough. For example, in Figure 5 and 6, there is no explanation in the figure or in the caption to tell what the meaning of the different lines and the numbers in legends are.

A: Changed to:

"*Figure 4. The relative uncertainty u'$_{rep}$ and mean concentration C$_{rep}$ for each set of repetitions (these repetitions include data from Isère, Colorado, Toutle, and Cowlitz Rivers (Spicer, 2019)). The relative uncertainty u'$_{nf}$ and the relative expanded uncertainty U'$_{nf}$ due to natural fluctuations correspond to the average and the average multiplied with a coverage factor of k = 2 of all tested relative uncertainties u'$_{rep}$.*

*Figure 5. Measured sand concentrations with uncertainty U'$_{meas}$ and exponential fits (represented as colored lines) for each sampling vertical using Bayesian modelling for the Rhône River at Lyon Perrache (a) Isère River at Grenoble Campus (b), Colorado River at River Mile 61 (c) and Amazon River at Manacapuru (d). The colors correspond to the sampling verticals, which are indicated by their distance in meters from the left bank, called abscissa.*

*Figure 6. Suspended-sand fluxes interpolated in each MAP cell along the sampling verticals (lines) and measured at the sampling points for the Rhône River at Lyon Perrache (a) Isère at Grenoble (b), Colorado River at River Mile 61 (c) and Amazon River at Manacapuru (d). The colors correspond to the sampling verticals, which are indicated by their distance in meters from the left bank, called abscissa.*"

4. Line 55. The VMT is mentioned several times in the paper without indicating its full form. I suggest to provide the full name of it in the introduction.

A: Included, thank you.

5. Similarly, for other abbreviations, e.g., please define BaM!" method in Line 168. MaxPost profile in Line 176. QRevInt in Line 57 etc., it would be nice to give full definition of them.

A: We included definitions of BaM! (Bayesian Modeling), QRevInt (Discharge Review International), MaxPost (best fitting, most probable profile) throughout the text.

6. Line 119-124. The choice of symbols u; u'; U; U' to represent different uncertainties is confusing. It is convention in fluid mechanics that u and U represent flow velocity; u' and U' represent velocity fluctuations. The author uses the velocity u in Equation (1) as well. I recommend use other symbols to represent these uncertainty metrics.

A: In metrology and uncertainty analysis, uncertainties and velocities are typically represented as $u$, so we had to make a choice on the use of $u$ for one of them. Since the article focuses on uncertainties, we decided to represent the uncertainties with their standard notion. We changed the use of $u$ in Equation (1) to $v$, and included $v$ into the notations at the end of the introduction.

7. Line 239. Another similar issue, h is defined as water depth in Line 92, while it is used to describe cell height here. It is better to use another symbol for cell height.
A: Thank you, we redefined the cell height as $h_{cell}$.

8. Line 224, please define what is hj.
A: Changed to: *"Thus, the ratio $C_{R,j}/h_j$ of the reference concentration $C_R$ and the water depth h for each column j in the MAP-grid is estimated through linear interpolation along the cross section."*

9. Line 119-124. A second suggestion on the terminologies. From description, here is my understanding, u is standard deviation σ (which is pointed out in the text). u' is the coefficient of variation CV. U is similar to kσ the z-score, etc. It will be helpful to include these common concepts in statistics in their definitions. My understanding might be correct or might be wrong. So please define clearly what are these terms (using equations) to avoid confusing.
A: This is true, but u/U is a common notation in metrology (uncertainty analysis). They are estimators of the real standard deviation σ and the covariance.

10. Line 187. In Eqn. (6), the constant should be $1.5 \times 10^3$.
A: Corrected, thank you.

**References**

Camenen B. & Larson M. (2007). A Unified Sediment Transport Formulation for Coastal Inlet Application. Technical Report ERDC/CHL CR-07-1, Coastal and Hydraulics Laboratory, USACE, Vicksburg, Mississippi, USA, 231p.

De Leeuw, J., Lamb, M. P., Parker, G., Moodie, A. J., Haught, D., Venditti, J. G., and Nittrouer, J. A. (2020). Entrainment and suspension of sand and gravel. Earth Surface Dynamics, 8(2):485-504.

Garcia, M. (2008). Sedimentation engineering: processes, measurements, modeling, and practice. American Society of Civil Engineers.

McLean, S. R. (1992). On the calculation of suspended load for noncohesive sediments. Journal of Geophysical Research , Vol. 97, No. C4 p. 5759-5770

Szupiany, R. N., Lopez-Weibel, C., Guerrero, M., Latosinski, F., Wood, M., Dominguez-Ruben, L. & Oberg, K. (2019). Estimating sand concentrations using ADCP-based acoustic inversion in a large fluvial system characterized by bi-modal suspended-sediment distributions. Earth Surface Processes & Landforms , Vol. 44, No. 6  p. 1295-1308

Venditti, J. G., Church, M.,  Attard, M. E. & Haught, D.  (02216). Use of ADCPs for suspended sediment transport monitoring:An empirical approach. Water Resources Research , Vol. 52
p. 2715-2736

Wright, S. and Parker, G. (2004). Flow resistance and suspended load in sand-bed rivers: simplifed stratification model. Journal of Hydraulic Engineering, 130(8):796-805.

---

## Author Response (AR2)

No changes made, manuscript accepted as it is.